# Parameter-Efficient Fine-Tuning for Large Models: A Comprehensive Survey

**Zeyu Han**                                                                                               *han.zeyu@northeastern.edu*

*Northeastern University*

**Chao Gao**                                                                                                       *cgao037@ucr.edu*
*University of California, Riverside*

**Jinyang Liu**                                                                                           *liu.jinyan@northeastern.edu*
*Northeastern University*

**Jeff (Jun) Zhang**                                                                                          *jeffzhang@asu.edu*
*Arizona State University*

**Sai Qian Zhang\***                                                                                          *sai.zhang@nyu.edu*
*New York University*

**Reviewed on OpenReview:** *https://openreview.net/forum?id=lIsCS8b6zj*

## Abstract

Large models represent a groundbreaking advancement in multiple application fields, enabling remarkable achievements across various tasks. However, their unprecedented scale comes with significant computational costs. These models, often consisting of billions of parameters, require vast amounts of computational resources for execution. Especially, the expansive scale and computational demands pose considerable challenges when customizing them for particular downstream tasks, particularly over the hardware platforms constrained by computational capabilities.

Parameter Efficient Fine-Tuning (PEFT) provides a practical solution by efficiently adjusting the large models over the various downstream tasks. In particular, PEFT refers to the process of adjusting the parameters of a pre-trained large model to adapt it to a specific task or domain while minimizing the number of additional parameters introduced or computational resources required. This approach is particularly important when dealing with large-scale language models with high parameter counts, as fine-tuning these models from scratch can be computationally expensive and resource-intensive, posing considerable challenges in the supporting system platform design.

In this survey, we present comprehensive studies of various PEFT algorithms, examining their performance and computational overhead. Moreover, we provide an overview of applications developed using different PEFT algorithms and discuss common techniques employed to mitigate PEFT computation costs. In addition to providing an extensive survey from an algorithmic standpoint, we also examine various real-world system designs to investigate the implementation costs associated with different PEFT approaches. This survey serves as a valuable resource for researchers aiming to understand both the PEFT algorithm and its system implementation, offering detailed insights into recent advancements and practical applications.

---

\*Corresponding author

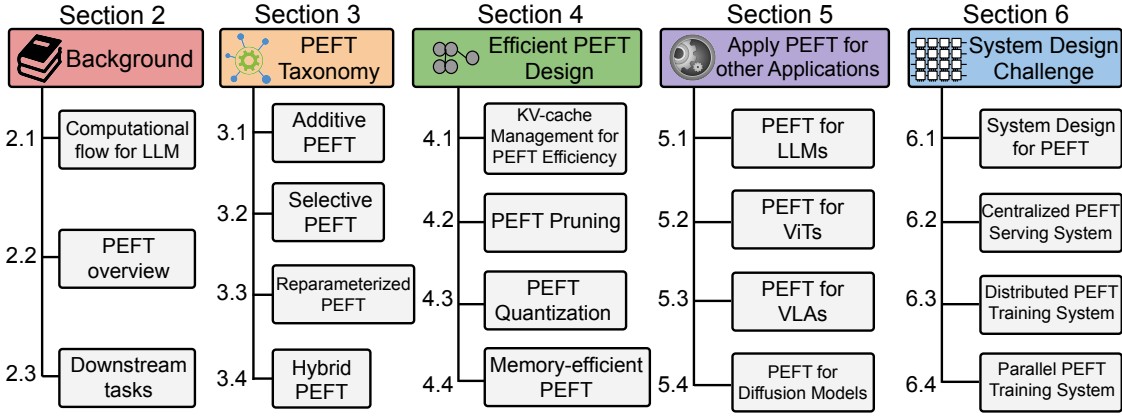

Figure 1: A content overview covered in the survey.

# 1 Introduction

Large Models (LMs) have recently captured considerable public interest. Their ability to understand context and nuances enables them to proficiently handle diverse tasks across multiple domains, including natural language processing (NLP), computer vision (CV), etc. In the field of NLP, Large Language Models (LLMs) have achieved significant advancements across various tasks including text generation (Brown et al., 2020; Zhuang et al., 2023), translation (Zhu et al., 2023c; Hadi et al., 2023), personalized chat-bots (Xu et al., 2023a; Li et al., 2023a; Wu et al., 2023c), and summarization (Zhang et al., 2023a), demonstrating remarkable proficiency.

Earlier studies (Brown et al., 2020) have suggested that LLMs exhibit high levels of generalization, enabling them to apply their acquired knowledge to new tasks not included in their original training. This capability is commonly known as *zero-shot learning*. Nevertheless, fine-tuning remains essential to further enhance LLMs for optimal performance on new user datasets and tasks.

Due to its scale, a widely adopted strategy for fine-tuning LLMs involves adjusting a limited number of LLM parameters while keeping the remainder unchanged. This technique, termed *Parameter-Efficient-Fine-Tuning* (PEFT), involves selectively adjusting a small proportion of their parameters while keeping the rest unaltered. Furthermore, the application of PEFT extends beyond the realm of NLP and quickly attracts interest in the CV community for handling fine-tuning vision models with large parameters, such as Vision Transformers (ViT) and diffusion models, as well as disciplinary models such as vision-language models.

In this survey, we systematically review and categorize recent advancements in PEFT algorithms as well as the system implementation costs associated with various PEFT algorithms across diverse scenarios. Figure 1 presents the overview content for this survey. In section 2, we present some fundamental concepts for LLM and PEFT, including computational flow for LLM, basic knowledge of PEFT, commonly used datasets and tasks, and evaluation benchmarks. We categorize all types of PEFT algorithms in Section 3 according to their computational flow. In Section 3.1, we detail additive algorithms that either introduce new weight parameters or modify activations. Algorithms that only require fine-tuning of existing parameters are categorized as selective approaches, which are introduced in Section 3.2. In Section 3.3, we explore reparameterized PEFT, which constructs a (low- dimensional) reparameterization of original model parameters for training while transforming the weights back to maintain the inference speed. Additionally, there exist algorithms that combine the above techniques, and we have classified these as hybrid approaches, elaborating on them in Section 3.4. We also investigate strategies for further reducing the computational complexity of different PEFT algorithms, including KV-cache management, pruning, quantization, and memory optimization, in Section 4.

In Section 5, we expand the scope of this survey beyond the computational perspective to involve various potential application scenarios. Specifically, we explore innovations that applying PEFT techniques to dif-

ferent model architecture, including LLMs (Section 5.1), Vision Transformer (Section 5.2), Vision-Language alignment models (Section 5.3), and Diffusion models (Section 5.4), for varied downstream tasks, underscoring PEFT's versatility and applicability in a range of scenarios. After that, in Section 6, we explore the system design challenge for PEFT methods. The discussion includes three advanced system solutions for practical PEFT deployment: PEFT query serving (Section 6.2), distributed tuning (Section 6.3), and concurrent PEFT tuning (Section 6.4). Finally, in Section 7, we summarize our survey and propose several potential future directions from both algorithmic and systemic perspectives, aiming to offer valuable insights for further research and development in the field.

## 2 Background

In this section, we first discussed the computation flow of LLM, including its fundamental components, computational complexity, and the flow of computations it involves as a case study. We then provide a brief overview of different PEFT algorithms in section 2.2.

### 2.1 Computation flow for LLaMA

In order to gain a deeper understanding of LLM and other Transformer-based models, we employ LLaMA-7B, a cutting-edge open-source LLM model, to scrutinize the architecture of LLM as well as Transformer. As shown in Figure 2 (a), LLaMA consists of three major components: an embedding block, a stack of decoder blocks, and a head block which consists of linear and softmax layers. The embedding layer's primary role is to transform unstructured textual information, into chunks of discrete numerical vectors (*tokens*) to facilitate subsequent processing. The embedded tokens are then delivered to the decoder layers for further processing. Each LLaMA decoder is composed of two fundamental components: Multi-head Self-Attention (MSA) and Feedforward Network (FFN). In the MSA module, each of the tokens will be clustered by an attention map obtained by a dot production between two linear mappings of the input tokens. Then the grouped tokens will be further processed by a Feedforward Neural network. Additionally, Root Mean Square Layer Normalization (RMSNorm) (Zhang & Sennrich, 2019) is adopted in LLaMA as a replacement for Layer Normalization to ensure efficient training.

LLM distinguishes itself from other deep neural network (DNN) models such as convolutional neural networks (CNN) in two significant ways. Firstly, LLM exhibits an inherent autoregressive nature, necessitating multiple iterations to complete the generation task. Moreover, LLM incorporates an attention mechanism, a component with computational complexity that scales quadratically with the length of the inputs. On the other hand, the inherent computation characteristic of LLM lies in the attention blocks inside each decoder layer. Figure 2 (c) depicts the high-level overview of the computation flow in the attention block.

During the inference process, each decoder takes a three-dimensional tensor $x \in \mathbb{R}^{b \times l \times d}$ as the input tokens. The input tokens are first multiplied with three weight matrices $W_Q$, $W_K$, and $W_V$, producing the output referred to as query($Q$), key($K$) and value($V$). Given the MSA module's inability to recognize positional data and the inherent auto-regressive nature of LLMs, the query and key will undergo a process using Rotary Positional Embedding (Su et al., 2021a) (RoPE, denoted as $R(.)$ in Eq 1) to encode the position information. Subsequently, the key and value will be combined with prior tokens.

After the positional embedding, the intermediate activation will then undergo a series of multiplication, softmax, and residual addition to generate MSA output as described in Eq 9. To be noted here, $d_k$ in the equation refers to the number of feature dimensions in the multi-head attention mechanism.

$$Q, K, V = R(W_q x), R(W_k x), W_v x \tag{1}$$

$$SA(x) = Softmax(\frac{QK^T}{\sqrt{d_{head}}})V \tag{2}$$

$$MSA(x) = [SA_1(x); SA_2(x); \dots; SA_k(x)]W_o \tag{3}$$

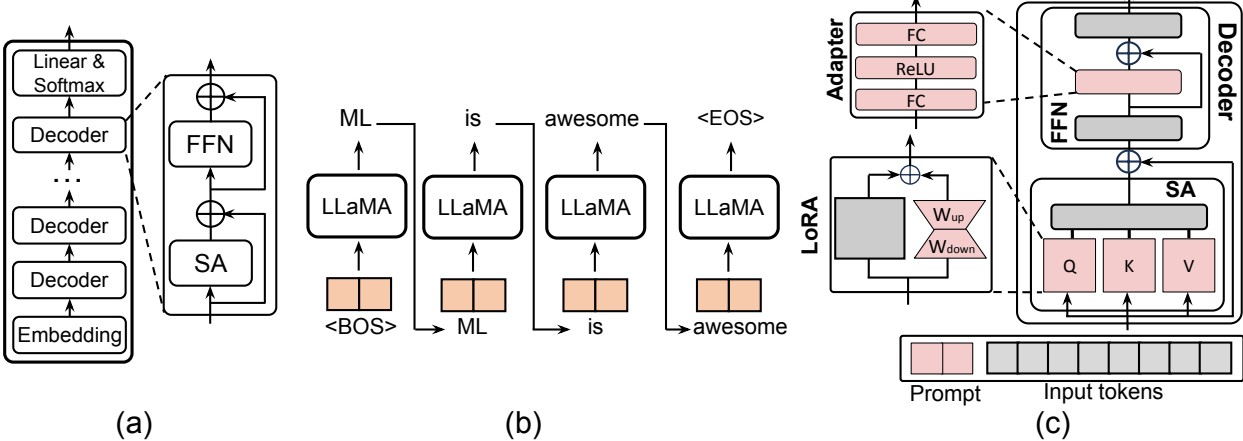

Figure 2: (a) LLaMA architecture. (b) LLaMA auto-regressive pattern. (c) Three common PEFT operations. All the learnable components are highlighted in red, while the frozen components are highlighted in grey. LoRA is applied on all the Query, Key, and Value blocks. The adapter targets the FFN module. Soft-Prompt focused on tuning the input activation of each decoder. We only show one decoder for illustration simplicity.

The SA output will then be forwarded to the FFN blocks for further processing. The FFN block will have another three matrices $W_{up}$, $W_{down}$, and $W_{gate}$ and the computation can be illustrated by:

$$FFN_{LLaMa}(x) = W_{up}(SiLU(W_{gate}x) \odot (W_{down}x)) + x, \tag{4}$$

where $x$ denotes the input of the FFN layer, and $SiLU$ is the nonlinear function used in LLaMA. In the original Transformer, the FFN block can be demonstrated by:

$$FFN_{Transfomer}(x) = W_{up}(ReLU(W_{down}x)) + x. \tag{5}$$

The output of the last decoder layer will be sent to a linear layer, which then generates a probability distribution spanning the complete *vocabulary* to predict the next token in the sequence. The produced token will then be concatenated with the previous tokens and used as the input for the next round of processing. This generating process repeats in an auto-regressive manner until a full sequence of tokens, referred to as a *completion*, is produced (Figure 2 (b)). For training, the computation flow is similar to that for inference, except that the generated sentences are directly compared to the ground truth output and generate the training loss. Gradients will then be computed across the LLM weights to minimize this training loss.

To analyze the computation cost and memory overhead in LLM, we also set a series of parameters used in later section 3. Table 1 shows the parameter size and computation dimension in the LLaMA-7B model as a starting example.

LLM models generate tokens (words) one for each round, depicted in Fig 2, based on the previous prompt (input) and previously generated sequence. This process will be repeated until the model outputs hits and termination token. To accelerate the inference process in LLM models, people take the strategy of storing the previous Keys and Values in the Key-Value cache (KV-cache), so they don't need to recalculate them for each new token. Mathematically, we can represent the total decoders' KV-cache memory cost in equation 6. In the equation, l and b are the context length and batch size and L refers to the number of layers. The $d_{head}$ is the head dimension and $n_{head}$ is the number of heads.

$$Size = L \times 2 \times b \times l \times d_{head} \times n_{head} \tag{6}$$

Table 1: Configuration parameters and computation operation for LLaMA-7B architecture

| Operation | Weights Symbol | Weights Dimension | Input Tensor Dimension | Complexity |
|:---:|:---:|:---:|:---:|:---:|
| Eq. 1 | $W_Q, W_K, W_V$ | $d \times k \times \frac{d}{k}$ | $b \times l \times d$ | $O(l)$ |
| Eq. 2 | - | - | $b \times l \times 3 \times k \times \frac{d}{k}$ | $O(l^2)$ |
| Eq. 3 | $W_o$ | $d \times d$ | $b \times l \times d$ | $O(l)$ |
| Eq. 4 | $W_{up}, W_{down}, W_{gate}$ | $d \times 4d$ | $b \times l \times d$  OR $l \times b \times 4d$ | $O(l)$ |

## 2.2 Overview on Parameter Efficient Fine Tuning

Fine-tuning remains essential to enhance LLM performance on unseen user datasets and tasks. With the size of the model growing (e.g. 1.5B in GPT-2 to 175B in GPT-3), standard full fine-tuning paradigm requires thousands of GPUs work in parallel, which is highly inefficient and unsustainable. A type of algorithm has been raised namely Parameter-efficient fine-tuning (PEFT) which aims to tune minimal parameters to achieve better performance over full tuning on downstream tasks.

In parallel developments, large-scale pre-trained models in vision and multimodal domains have also demonstrated their effective representational learning capabilities, enabling adaptation from large datasets to smaller ones or across various data modalities through fine-tuning. Consequently, this capability has made PEFT increasingly attractive to the wider research community.

We categorized the PEFT algorithms into **additive, selective, reparameterized, and hybrid** fine-tuning based on their operations. As Figure 3 depicts, three major **additive fine-tuning** algorithms are normally used: (1) Adapter; (2) Soft Prompt; (3) Others. They differ in terms of the additional tunable modules or parameters. **Selective fine-tuning**, on the other hand, doesn't require any additional parameters, it selects a small subset of parameters from the backbone model and only makes them tunable while keeping the majority of parameters untouched during fine-tuning on downstream tasks. We categorized selective fine-tuning based on the grouping of chosen parameters: (1) Unstructural Masking; and (2) Structural Masking. Reparametrization represents transforming model parameters between two equivalent forms. Specifically, **reparametrized fine-tuning** introduces additional low-rank trainable parameters during training, which are then integrated with the original model for inference. This approach is categorized into two main strategies: (1) Low-rank Decomposition, and (2) LoRA Derivatives. **Hybrid fine-tuning** explores the design spaces of different PEFT methods and combines their advantages.

## 2.3 Downstream Tasks for LLM Evaluation

Two types of tasks have been widely used for LLM evaluation, the first type is the General Language Understanding Evaluation (GLUE) (Wang et al., 2018) benchmark, which integrates nine sentence or sentence-pair language understanding tasks (CoLA, SST-2, MRPC, STS-B, QQP, MNLI, QNLI, RTE, and WNLI), chosen for their diversity in dataset sizes, text genres, and difficulty levels, and is based on established existing datasets. It also includes a diagnostic dataset designed to evaluate and analyze model performance across diverse linguistic phenomena in natural language. Additionally, it features a public leaderboard to track performance on the benchmark and a dashboard to visualize model performance on the diagnostic set.

The other type of dataset that has been used in recent LLM papers is common sense reasoning which integrated into our study caters to a variety of research facets: (1) *OpenBookQA* (Mihaylov et al., 2018) is curated to foster research in advanced question-answering, delving into a profound understanding of both the subject matter and the language in which it is articulated. (2) *PIQA* (Bisk et al., 2020) primarily emphasizes everyday scenarios, demonstrating a predilection for unconventional solutions. (3) *Social IQA* (Sap et al., 2019) emerges as a novel question-answering benchmark tailored for gauging social commonsense intelligence. (4) *HellaSwag* (Zellers et al., 2019) serves as a dataset, the essence of which is to ascertain the capability of machines in aptly concluding sentences. (5) *BoolQ* (Clark, 2019) is a dataset dedicated to question-answering, particularly for binary responses (yes/no queries). (6) *WinoGrande* (Sakaguchi et al., 2021) is introduced as a fresh compilation, encompassing a substantial 44,000 problems. (7) *ARC-easy* (Clark et al., 2018) presents

itself as a novel dataset constituting genuine grade-school level multiple-choice science questions, designed to invigorate research in intricate question-answering. (8) *ARC-challenges* (Clark et al., 2018), distinctively, encompasses solely those questions that were inaccurately addressed by both a retrieval-based algorithm and a word co-occurrence algorithm.

Image recognition serves as a key benchmark and application for vision models, illustrated by tasks like fine-grained visual categorization (FGVC) and the visual task adaptation benchmark (VTAB). Beyond image classification, video action recognition is another key application area, involving datasets like Kinetics-400 (Kay et al., 2017), SSv2 (Goyal et al., 2017), and HMDB51 (Kuehne et al., 2011). Additionally, PEFT has been utilized for dense prediction tasks, using datasets like MSCOCO (Lin et al., 2014), ADE20K (Zhou et al., 2017), and PASCAL VOC (Everingham et al., 2010).

## 2.4 Evaluation Benchmarks for PEFT

A comprehensive benchmark is essential for readers to evaluate performance differences among various PEFT methods under a unified standard. We next discuss several commonly used benchmarks.

From the algorithmic perspective, (Ding et al., 2023b) benchmarks the performance of several PEFT algorithms across more than 100 NLP tasks and conducts systematic experiments based on criteria such as performance, convergence, efficiency, combinability, scalability, and transferability. Similarly, (Xu et al., 2023b) and (Pu et al., 2023) have also established targeted benchmarks to evaluate different PEFT algorithms.

From the system perspective, three commonly used benchmarks are outlined below to evaluate system performance. The first benchmark is the ShareGPT dataset (OpenAI, 2023a), which includes real-world interactions with OpenAI's ChatGPT. It encompasses a broad spectrum of conversational queries and responses that are representative of typical user interactions with large language models (LLMs). This dataset is vital for evaluating the system's ability to manage diverse and realistic conversational requirements, focusing on the accuracy of responses and efficiency in handling requests.

The second benchmark involves the Microsoft Azure Function Trace from the years 2019 and 2021 (Microsoft, 2023), containing logs from serverless computing activities via Azure Functions. While these logs are from a general serverless computing context rather than LLM-specific applications, they offer insights into the computational demands driven by events. These traces simulate the arrival patterns and workload intensities that LLM systems might face, including irregular and peak demands, thus acting as practical proxies for LLM inference tasks.

The third benchmark is based on the Gamma process (Moreno et al., 2014), a prevalent approach in simulations to model the timing of incoming requests in queueing and service systems. This method facilitates the creation of workloads with varied arrival rates and patterns, producing synthetic, yet realistic request scenarios that a system could encounter during actual operations. Such synthetic workloads are crucial for testing system performance under controlled conditions that resemble real-world user activity.

## 3 PEFT Taxonomy

The PEFT strategies can be broadly classified into four categories: **additive PEFT** (Section 3.1), which modifies the model architecture by injecting new trainable modules or parameters; **selective PEFT** (Section 3.2), which makes a subset of parameters trainable during fine-tuning; **reparameterized PEFT** (Section 3.3), which constructs a (low-dimensional) reparameterization of the original model parameters for training, then equivalently transforms it back for inference; and **hybrid PEFT** (Section 3.4), which combines advantages from different PEFT methods to build a unified PEFT model. An overview of different types of PEFT algorithms is depicted in Figure 4.

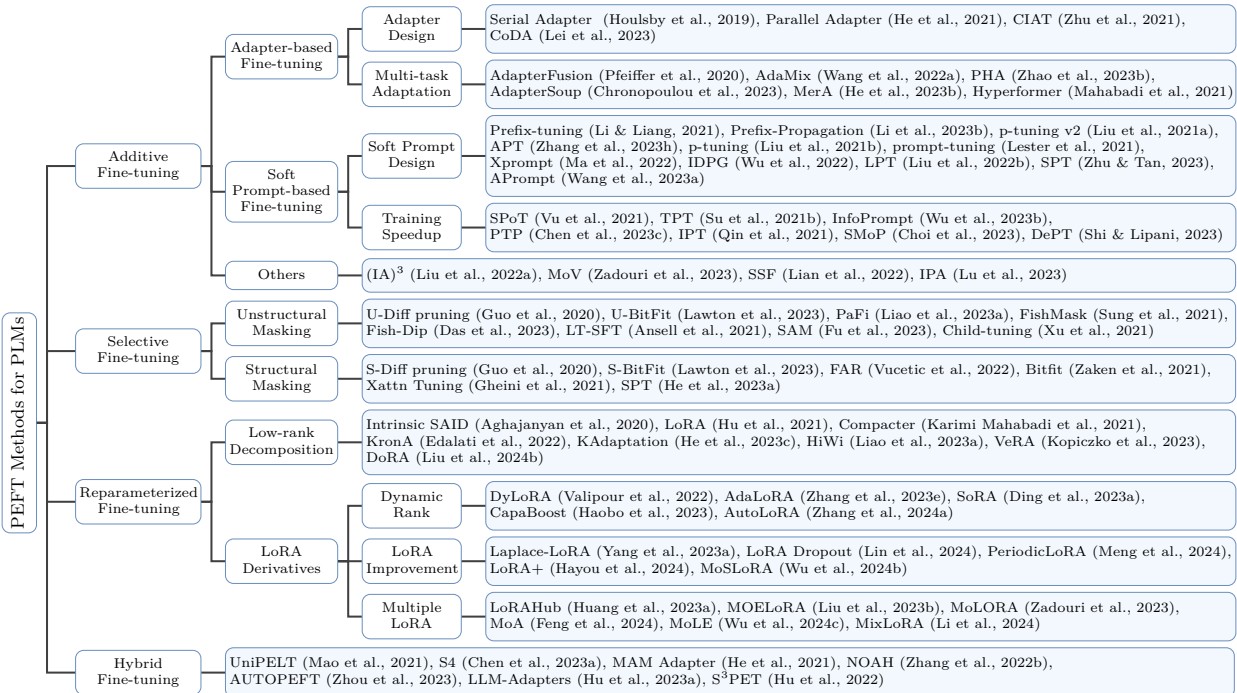

Figure 3: Taxonomy of Parameter-Efficient Fine-Tuning Methods for Large Models.

(a) Additive PEFT    (b) Selective PEFT    (c) Reparameterization PEFT

Figure 4: Different types of PEFT algorithms.

## 3.1 Additive PEFT

Standard full fine-tuning entails substantial computational expenses and could also potentially harm the model's generalization ability. To mitigate this problem, a widely employed approach is to maintain the pre-trained backbone unchanged and introduce only a minimal number of trainable parameters that are strategically positioned within the model architecture. While fine-tuning for a specific downstream task, only the weights of these additional modules or parameters are updated, which results in a substantial reduction in storage, memory, and computational resource requirements. Due to their characteristic of adding parameters, these techniques can be termed as *Additive Tuning*, as shown in Figure 4 (a). Next, we discuss several popular Additive PEFT algorithms.

### 3.1.1 Adapters

Adapter approaches involve the insertion of small adapter layers within Transformer blocks. Typically, an adapter layer consists of a down-projection matrix $W_{\text{down}} \in \mathbb{R}^{r \times d}$, followed by a non-linear activation function $\sigma(\cdot)$, and an up-projection matrix $W_{\text{up}} \in \mathbb{R}^{d \times r}$. In this context, $d$ represents the dimension of the hidden layer, and $r$ serves as the bottleneck dimension, which is a hyperparameter used in configuring the adapters.

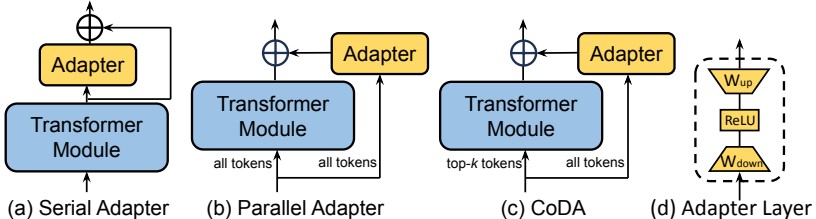

Figure 5: Illustration of three representative adapter-based fine-tuning algorithms. Blue represents frozen, while yellow represents trainable.

Denote $h_{in}$ as the input to the adapter, the computation within the adapter module (with residual) can be summarized as follows:

$$Adapter(x) = W_{\text{up}}\sigma(W_{\text{down}}x) + x. \tag{7}$$

The concept of adapters in the field of NLP was initially introduced by **Serial Adapter** (Houlsby et al., 2019) as shown in Figure 5 (a). In their approach, each Transformer block is enhanced by adding two adapter modules, with one positioned after the self-attention layer and the other after the FFN layer, respectively. Subsequent research has aimed to address the additional computational cost associated with adapter layers. A modified framework **AdapterFusion** (Pfeiffer et al., 2020) was proposed, where adapter layers are inserted only after the 'Add & Norm' step following the FFN layer to enhance the computational efficiency. The adapters mentioned above follow a sequential design, placing adapter layers as bottlenecks within the Transformer blocks. This approach may potentially reduce the model's parallelism and require a trade-off between inference efficiency and accuracy. In contrast, He et al. (2021) introduced a **parallel adapter** (PA) approach as depicted in Figure 5 (b), which reorganizes the traditionally sequential adapter layers into a parallel side-network that runs alongside each Transformer sublayer. Similarly, **CIAT** (Zhu et al., 2021), **CoDA** (Lei et al., 2023) and **KronA** (Edalati et al., 2022) also adopts a parallel adapter design. Except for the parallel design, **CoDA** employs a sparse activation mechanism to improve the inference efficiency as shown in Figure 5 (c). Specifically, CoDA uses a soft top-$k$ selection process that identifies $k$ important tokens in each layer, which will be processed by both the frozen pre-trained Transformer layer and the adapter branch to maintain model accuracy. In contrast, those unimportant tokens are only processed by the adapter branch while skipping the heavy pre-trained layer, therefore optimizing for inference efficiency without compromising overall performance.

To enhance the performance and generalization of adapters, various studies have implemented *multi-task learning* strategies, such as **AdapterFusion** (Pfeiffer et al., 2020), **AdaMix** (Wang et al., 2022a), **PHA** (Zhao et al., 2023b), **AdapterSoup** (Chronopoulou et al., 2023), **MerA** (He et al., 2023b), and **Hyperformer** (Mahabadi et al., 2021). **AdapterFusion** keeps all pre-trained adapters in the model and employs a fusion module to merge the multi-task information. Unlike AdapterFusion, **MerA** merges pre-trained adapters into a single one through optimal transport based on weights and activations. This approach avoids introducing any additional trainable parameters, thereby enhancing computational efficiency. **Hyperformer** stores the multi-task information in a shared hypernetwork, which generates task and layer-specific adapter parameters conditioned on task and layer ID embeddings. Given a new task, only an additional task embedding needs to be learned, therefore reducing the number of trained parameters.

### 3.1.2 Soft Prompt

Alternatively, prompt tuning presents an additional approach for refining the model to achieve improved performance through fine-tuning. Instead of optimizing discrete token representations through in-context learning, there is a prevailing belief that the continuous embedding space of soft prompts inherently contains more information (Petrov et al., 2023). Drawing inspiration from this concept, researchers directly prepend adjustable vectors, referred to as soft prompts, to the start of the input sequence. This can be represented as follows:

$$\mathbf{X}^{(l)} = [\mathbf{s}_1^{(l)}, \ldots, \mathbf{s}_{N_S}^{(l)}, \mathbf{x}_1^{(l)}, \ldots, \mathbf{x}_{N_X}^{(l)}] \tag{8}$$

where $\mathbf{X}^{(l)}$ is the sequence of input tokens for layer $l$, including soft prompt tokens $\mathbf{s}_i^{(l)}$ followed by the original input tokens $\mathbf{x}_i^{(l)}$. $N_S$ is the number of soft prompt tokens, and $N_X$ is the number of original input tokens.

**Prefix-tuning** (Li & Liang, 2021) introduces learnable vectors that are prepended to keys $k$ and values $v$ across all Transformer layers. To ensure stability during the optimization process, Prefix-tuning adopts a reparameterization strategy, which utilizes an MLP layer to generate these prefix vectors rather than optimizing them directly. After fine-tuning, only the prefix vectors are saved for inference. This technique has been adapted and improved in several studies (Li et al., 2023b; Liu et al., 2021a; Zhang et al., 2023h). For instance, **p-tuning v2** (Liu et al., 2021a) removes reparameterization and expands its usage to broader model scales and NLP tasks. **APT** (Adaptive Prefix Tuning) (Zhang et al., 2023h) enhances Prefix-tuning by introducing an adaptive gate mechanism to control the prefix importance in each layer. Concurrent work **p-tuning** (Liu et al., 2021b) and **prompt-tuning** (Lester et al., 2021) apply learnable vectors only at the initial word embedding layer rather than all layers to enhance training and inference efficiency. It's important to highlight that prompt-tuning demonstrates its effectiveness primarily in the context of large models, specifically those with over 11 billion parameters (Lester et al., 2021). Complementing this, **Xprompt** (Ma et al., 2022) eliminates the negative prompt tokens through a hierarchically structured pruning, which closes the performance gap at smaller model scales. **Wang et al. (2023c)** provides some theoretical analysis towards prompt tuning, demonstrating its universality and limitations in limited-depth Transformers. **IDPG** (Instance-Dependent Prompt Generation) (Wu et al., 2022) improves prompt tuning by generating prompts based on each input sentence with a lightweight prompt generator. In a related approach, **LPT** (Late Prompt Tuning) (Liu et al., 2022b) also leverages a prompt generator to obtain instance-aware prompt. Unlike previous work, LPT adds these prompts only after an intermediate layer, rather than at the initial or all layers. This strategic placement eliminates the gradient calculation below the intermediate layer, thereby significantly accelerating the training speed. Simultaneously, LPT can improve the overall performance due to the shorter backpropagation path preserves more task-related information. Inspired by LPT, **SPT** (Selective Prompt Tuning) (Zhu & Tan, 2023) delves deeper into the importance of prompt inserting strategies. It introduces a learnable probabilistic gate in each layer to determine whether to use the prompt propagated from the previous layer or inject a newly generated prompt. **APrompt** (Wang et al., 2023a) employs another prompt inserting strategy. In addition to input prompts inserted at the beginning of the input sequence for each Transformer layer, APrompt also prepends additional learnable prompts to the respective query, key, and value matrices in the self-attention blocks to learn new attention patterns. Besides, APrompt incorporates the learning of a task-specific head.

The concept of soft prompts has been employed for various downstream tasks (Choi & Lee, 2023; Wu & Shi, 2022), although their training can be prone to instability and slow convergence. To address this, **SPoT** (Vu et al., 2021) uses a source prompt learned from one or multiple tasks to initialize prompts for new tasks. Similarly, the transfer of soft prompts from one task to initialize another is proposed in **TPT** (transferable prompt tuning) (Su et al., 2021b), which demonstrates that a better prompt initialization results in a large training convergence speedup. **InfoPrompt** (Wu et al., 2023b) develops two mutual information-based loss functions, i.e., *head loss* and *representation loss*, to find better prompt initialization and learn sufficient task-relevant information, thereby also expediting convergence. **PTP** (Chen et al., 2023c) delves into the root causes of training instability. It identifies the steep nature of the loss landscape in conventional prompt tuning, where minor variations in input data can lead to significant loss fluctuations. To mitigate this, PTP introduces perturbation-based regularizers to smooth the loss landscape and consequently stabilize the training process. **DePT** (Shi & Lipani, 2023) decomposes the soft prompt into a shorter soft prompt with a pair of low-rank matrices, which are optimized with two distinct learning rates. This strategy not only improves performance but also enhances training and inference efficiency. **SMoP** (Sparse Mixture-of-Prompts) (Choi et al., 2023) reduce the training and inference cost by utilizing short soft prompts. During training, multiple short soft prompts are trained, each tailored to specific subsets of the dataset. During inference, SMoP integrates a gating mechanism that routes each input instance to an appropriate short prompt. This technique not only increases efficiency in both training and inference stages but also retains

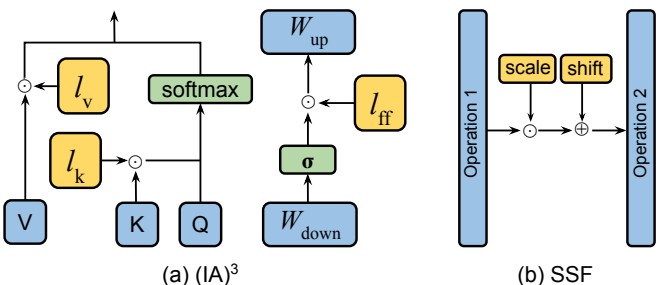

Figure 6: Illustration of $(IA)^3$ and SSF. Blue represents frozen, while yellow represents trainable.

performance comparable to those achieved with longer soft prompts. To further cut down the number of soft prompt parameters, **IPT** (Intrinsic Prompt Tuning) (Qin et al., 2021) identifies an intrinsic task subspace by training an auto-encoder on multiple tasks. Tuning on new tasks then requires adjusting only a few parameters within this subspace, significantly reducing the number of training parameters.

### 3.1.3 Other Additive Methods

Apart from the methods mentioned above, there appear other approaches that strategically incorporate additional parameters during the fine-tuning process. For example, $(IA)^3$ (Liu et al., 2022a) introduces three learnable rescaling vectors: $l_k \in \mathbb{R}^{d_k}$, $l_v \in \mathbb{R}^{d_v}$, and $l_{ff} \in \mathbb{R}^{d_{ff}}$, to rescale the key, value, and FFN activations, respectively, as depicted in Figure 6 (a). The operations within the self-attention block can be described as follows:

$$SA(x) = Softmax(\frac{Q(l_k \odot K^T)}{\sqrt{d_{head}}})((l_v \odot V). \tag{9}$$

In FFN, the rescaling can be denoted as:

$$FFN_{Transfomer}(x) = W_{up}(l_{ff} \odot \sigma(W_{down}x)), \tag{10}$$

where $\odot$ is Hadamard product. Furthermore, the scale vectors $l_k$ and $l_v$ can be seamlessly integrated into the weight matrices of $A_Q$ and $A_W$. This integration effectively eliminates the extra computational costs during inference. A similar technique **SSF** (Lian et al., 2022) also performs linear transformation to the model activations, as illustrated in Figure 6 (b). Specifically, after each operation (i.e., MSA, FFN, and layer normalization) in the pre-trained model, an SSF-ADA layer is injected, which performs scaling and shifting to the features generated from the operation. During fine-tuning, only those SSF-ADA layers can be updated, while during inference, similar to $(IA)^3$, these SSF-ADA layers can be merged into model weights, so no additional inference overhead would be incurred. **IPA** (Inference-Time Policy Adapters) (Lu et al., 2023) offers a novel approach to align LLMs, such as GPT-4, with user-specific requirements without modifying the base model's parameters. This is particularly significant when dealing with models whose parameters are extremely large and often not directly accessible. IPA achieves this by combining (through multiplication and normalization) the output distribution of a base LLM (base policy) with that of a smaller-sized model (adapter policy) during the decoding phase. During training, the policy adapter's parameters are fine-tuned using reinforcement learning, while the base policy's parameters remain fixed. During inference, IPA decodes with the combined distribution of the base model and the trained policy adapter, tailoring it to fulfill specific user-defined criteria.

## 3.2 Selective PEFT

Rather than additive PEFT, which increases the model complexity by adding more parameters, selective PEFT fine-tunes a subset of the existing parameters to enhance model performance over downstream tasks, as depicted in Figure 4 (b).

Specifically, given a model with parameters $\theta = \{\theta_1, \theta_2, ..., \theta_n\}$ where each $\theta_i$ denotes an individual model parameter and $n$ represents the total count of these parameters, the process of selective PEFT is represented

by applying a binary mask $M = \{m_1, m_2, ..., m_n\}$ to these parameters. Each $m_i$ in $M$ is either 0 or 1, indicating whether the corresponding parameter $\theta_i$ is selected (1) or not selected (0) for fine-tuning. The updated parameter set $\theta'$ after fine-tuning is given by:

$$\theta_i' = \theta_i - \eta \cdot m_i \cdot \frac{\partial \mathcal{L}}{\partial \theta_i} \tag{11}$$

where $\eta$ represents the learning rate, and $\frac{\partial \mathcal{L}}{\partial \theta_i}$ is the gradient of the loss function with respect to the parameter $\theta_i$. In this formulation, only the selected parameters (i.e., $m_i = 1$) are updated during backpropagation.

**Diff pruning** (Guo et al., 2020) is a representative work that applies a learnable binary mask to the model weights during fine-tuning. To achieve parameter efficiency, the mask is regularized by a differentiable approximation of the $L_0$-norm penalty. **PaFi** (Liao et al., 2023a) simply select model parameters with the smallest absolute magnitude as trainable. **FishMask** (Sung et al., 2021) determines parameter importance using the approximate Fisher information. It then selects the top k parameters based on this information to form the mask $M$. Similarly, **Fish-Dip** (Das et al., 2023) also uses Fisher information to calculate $M$, but the mask will be re-calculated dynamically in each train period. **LT-SFT** (Ansell et al., 2021) introduces another technique to determine parameter importance inspired by the Lottery Ticket Hypothesis (Frankle & Carbin, 2018; Malach et al., 2020), where the subset of parameters that change the most during an initial fine-tuning stage is selected to form the mask $M$. **SAM** (Fu et al., 2023) proposes a second-order approximation method, which approximates the original problem with an analytically solvable optimization function, to help decide the parameter mask. **Child-tuning** (Xu et al., 2021) proposes two approaches to select a child network during each training iteration, where only the parameters within this child network can be updated.

However, the above unstructured parameter masking results in an uneven distribution of non-zero masks and diminished hardware efficiency when implementing PEFT. As shown in Figure 7, the structured mask organizes parameter masking in regular patterns, unlike unstructured ones that apply it randomly, thus enhancing computational and hardware efficiency during training. Therefore, various structured selective PEFT techniques have undergone extensive investigation. **Diff pruning** proposes a structured pruning strategy by partitioning the weight parameters into local groups and strategically eliminating them together. Similarly, **FAR** (Vucetic et al., 2022) fine-tunes BERT models by grouping weights of the FFN in Transformer blocks into nodes, then ranking and selecting the learner nodes using $L_1$ norm. To further reduce the memory access frequency, they also reconfigure the FFN by grouping the learner nodes.

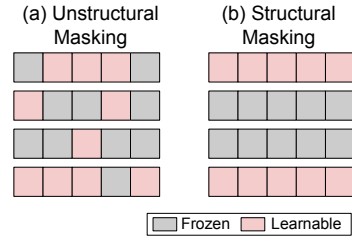

Figure 7: Illustration of two parameter masking methods.

**Bitfit** (Zaken et al., 2021) is proposed to only fine-tune the bias parameters of each DNN layer, and achieve competitive results for small models. However, this method fails to handle large models. **Lawton et al. (2023)** applies NAS to Bitfit, where **S-BitFit** keeps the structural nature in Bitfit that restricts NAS algorithm must choose whether $\delta b = 0$ or not for each bias module. Similar to Bitfit fine-tunes a specific module in Transformer, **Xattn Tuning** (Gheini et al., 2021) fine-tunes only the cross-attention layers. **SPT** (sensitivity-aware visual parameter-efficient fine-tuning) (He et al., 2023a) first identifies the sensitive parameters measured by the loss reduction when being tuned. This sensitivity is calculated using a first-order Taylor expansion, derived from a single forward and backward pass before fine-tuning in one shot. Next, SPT finds the weight matrices whose number of sensitive parameters exceeds a pre-defined threshold and then applies a selected PEFT technique (e.g., LoRA and Adapter) to these targeted weights to achieve structural tuning.

### 3.3 Reparameterized PEFT

Reparameterization stands for equivalently transforming a model's architecture from one to another via transforming its parameters. In the context of PEFT, this often means constructing a low-rank parameterization to achieve the goal of parameter efficiency during training. For inference, the model can be converted

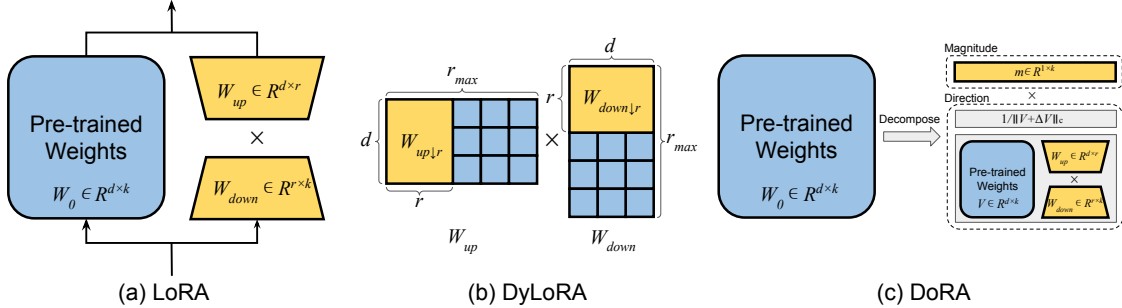

Figure 8: Illustration of three representative reparameterized PEFT algorithms. Blue represents frozen, while yellow represents trainable.

to its original weight parameterization, ensuring unchanged inference speed. This procedure is depicted in Figure 4 (c).

Earlier research studies (Aghajanyan et al., 2020) have shown that common pre-trained models exhibit an exceptionally low intrinsic dimensionality. In other words, it is possible to find a low-dimensional reparameterization that is effective for fine-tuning as the entire parameter space. **Intrinsic SAID** (Aghajanyan et al., 2020) is the pioneering work in investigating the intrinsic dimension feature during the fine-tuning of LLMs. However, the most widely recognized reparameterization technique is **LoRA** (Low-Rank Adaptation) (Hu et al., 2021; Fomenko et al., 2024), as shown in Figure 8 (a). For a given pre-trained weight matrix $W_0 \in \mathbb{R}^{d \times k}$, LoRA introduces two trainable weight matrices, $W_{\text{up}} \in \mathbb{R}^{d \times r}$ and $W_{\text{down}} \in \mathbb{R}^{r \times k}$ where the rank $r \ll min(d, k)$, operating in parallel to $W_0$. Let $h_{in}$ represent the input. Under normal conditions, the output through $W_0$ is $h_{out} = W_0 h_{in}$. Instead, LoRA modifies this output by introducing an incremental update $\Delta W$ that encapsulates task-specific knowledge:

$$h_{out} = W_0 h_{in} + \frac{\alpha}{r} \Delta W h_{in} = W_0 h_{in} + \frac{\alpha}{r} W_{\text{up}} W_{\text{down}} h_{in}, \tag{12}$$

where $\alpha$ denotes a scaling factor. At the onset of training, $W_{\text{down}}$ is initialized using a random Gaussian distribution, while $W_{\text{up}}$ is initialized to zero, ensuring that $\Delta W$ initially holds a value of zero. LoRA is straightforward to implement and has been evaluated on models with up to 175 billion parameters. Fig 8 (c) used a single decoder as an example, the frozen and learnable components are highlighted in grey and red, respectively. Once fine-tuning is complete, LoRA's adaptive weights seamlessly integrate with the pre-trained backbone weights. This integration ensures that LoRA maintains the model's efficiency, adding no extra burden during inference.

In LoRA training, selecting an appropriate rank has always been a challenging issue. To address this, **DyLoRA** (Valipour et al., 2022), as depicted in Figure 8 (b), trains the LoRA module on a range of ranks within a predefined training budget, rather than adhering to a single, fixed rank. Specifically, for a given rank range $R = \{r_{\min}, r_{\min} + 1, \ldots, r_{\max}\}$, DyLoRA dynamically chooses a rank $r \in R$ at each iteration of the training process. Consequently, the matrices $W_{\text{down}}$ and $W_{\text{up}}$ are tailored for the selected rank $r$, resulting in truncated versions $W_{\text{down}\downarrow r} = W_{\text{down}}[1 : r, :]$ and $W_{\text{up}\downarrow r} = W_{\text{up}}[:, 1 : r]$, and the subsequent forward and backward pass during this iteration will be restricted on $W_{\text{down}\downarrow r}$ and $W_{\text{up}\downarrow r}$ instead of $W_{\text{down}}$ and $W_{\text{up}}$. With this dynamic and search-free approach, DyLoRA significantly reduces the training time required to find an optimal and fixed LoRA rank for specific tasks. **AdaLoRA** (Zhang et al., 2023e) reformulates the $\Delta W$ with a singular value decomposition (SVD), denoted as $\Delta W = P \Lambda Q$, where $P \in \mathbb{R}^{d \times r}$ and $Q \in \mathbb{R}^{r \times k}$ are orthometric, $\Lambda$ is a diagonal matrix containing singular values $\{\lambda_i\}_{1 \leqslant i \leqslant r}$. All three weight matrices are made learnable. During training, the singular values are pruned iteratively based on their importance scores, which are constructed from the moving average of the magnitude of the gradient-weight product. To ensure the orthogonality between $P$ and $Q$, i.e., $P^T P = Q Q^T = I$, an additional regularizer term is included in the loss:

$$R(P, Q) = \left\| P^T P - I \right\|_F^2 + \left\| Q Q^T - I \right\|_F^2. \tag{13}$$

This adaptive approach enables the model to dynamically adjust the rank within each LoRA module, effectively managing its parameter counts based on the significance of the weight matrices. However, according to **SoRA** (Ding et al., 2023a), the importance scores used in AdaLoRA are heuristically constructed, which lacks rigorous theoretical motivation. Additionally, both moving average operation and calculation of Eq. 13 introduce extra computation costs during training. To address this, SoRA eliminates the orthogonality premise of $P$ and $Q$. Instead, a gating unit $g \in \mathbb{R}^r$ between $W_{\text{up}}$ and $W_{\text{down}}$ is directly applied and optimized:

$$h_{out} = W_{\text{up}}(g \odot (W_{\text{down}} h_{in})), \tag{14}$$

where $\odot$ is Hadamard product. The gate $g$ is updated using a variation of proximal gradient iteration for $l_1$ loss (Beck & Teboulle, 2009; Chambolle et al., 1998), which has a clear mathematical meaning and does not need the heuristic premise. After training, the zeroed-out gate units are pruned by removing the corresponding columns and rows in $W_{\text{down}}$ and $W_{\text{up}}$.

Several subsequent studies have aimed to improve LoRA's performance in various aspects. For instance, **Laplace-LoRA** (Yang et al., 2023a) notices that fine-tuned LLMs often exhibit overconfidence. To enhance the calibration of fine-tuned LLMs, Laplace-LoRA utilizes a Bayesian approach, specifically a post-hoc Laplace approximation (MacKay, 1992; Antorán et al., 2022), to the posterior over the LoRA parameters. **LoRA Dropout** (Lin et al., 2024) introduces random noises to the learnable low-rank matrices and increases parameter sparsity to reduce the risk of overfitting. **LoRA+** (Hayou et al., 2024) proposes to set different learning rates for the LoRA matrices $W_{\text{down}}$ and $W_{\text{up}}$, such that $\eta_{\text{up}} = \lambda \eta_{\text{down}}$ with $\lambda > 1$ fixed and tune $\eta_{\text{down}}$. **MoSLoRA** (Mixture-of-Subspaces LoRA) (Wu et al., 2024b) decomposes LoRA into subspaces via structural reparameterization, then employs a learnable mixer, trained jointly with the original LoRA weights, to fuse the subspaces. Similarly to LoRA, MoSLoRA can also be merged into the original weights.

Thanks to the modular design of LoRA, many studies incorporate multiple LoRA modules in their frameworks to enhance performance. For example, **LoRAHub** aggregates various LoRA modules trained on different tasks. Given a handful of examples from a new task, LoRAHub can autonomously compose compatible LoRA modules without human intervention via a gradient-free method Shiwa (Liu et al., 2020). **MOELoRA** employs a Mixture-of-Experts (MOE) approach to train LoRA in a multi-task setting, resulting in multiple expert LoRA modules. To retrieve parameters for certain tasks, MOELoRA utilizes a task-motivated gate function that assigns contribution weights to each expert based on the task ID, and the final parameters are calculated through a weighted sum of all experts.

In addition to LoRA, several other reparameterization techniques are emerging with significant potential. For instance, **Compacter** (Karimi Mahabadi et al., 2021) introduces a light-weight adapter modules by parameterizing the $W_{\text{down}}$ and $W_{\text{up}}$ as $W = \sum_{i=1}^{n} A_i \otimes B_i$, where $A_i \in \mathbb{R}^{n \times n}$, $B_i \in \mathbb{R}^{\frac{r}{n} \times \frac{d}{n}}$, and $\otimes$ denotes the Kronecker product. They further decrease the parameter count by designating $A_i$ as shared parameters and reparameterizing $B_i$ using the product of two low-rank matrices, effectively reducing the parameter complexity from $\mathcal{O}(rd)$ to $\mathcal{O}(r+d)$. Related studies, such as **KronA** (Edalati et al., 2022) and **KAdaptation** (He et al., 2023c), also employ the Kronecker product to reparameterize adapter weights, aiming to achieve parameter reduction. **HiWi** (Liao et al., 2023a) proposes an adapter fine-tuning method that applies an adapter directly to pre-trained parameters instead of hidden representations as:

$$W' = W + \sigma(W W_{\text{down}}) W_{\text{up}}, \tag{15}$$

where $W$ denotes the weights or biases within the Transformer block's feed-forward layer. Notably, during inference, this method computes $W'$ in advance, ensuring that the model's inference latency remains on par with that of traditional full fine-tuning. **VeRA** (Vector-based Random Matrix Adaptation) (Kopiczko et al., 2023) employs a single pair of frozen low-rank matrices $W_{\text{up}}$ and $W_{\text{down}}$ that are shared across all layers, and adapts these matrices by learning small, trainable scaling vectors represented as $b$ and $d$ (formally denoted by diagonal matrices $\Lambda_b$ and $\Lambda_d$). Specifically, the reparameterization is given by:

$$h_{out} = W_0 h_{in} + \Lambda_b W_{\text{up}} \Lambda_d W_{\text{down}} h_{in}, \tag{16}$$

where both $W_{\text{up}}$ and $W_{\text{down}}$ are initialized using a random Gaussian distribution. Similar to LoRA, the scaling vector $b$ is initialized to zeros to ensure that the weight matrix is unaffected during the first forward pass. This

method significantly reduces the number of trainable parameters compared to LoRA yet maintains the same performance, enabling the fine-tuning of larger models on a single GPU. **DoRA** (Weight-Decomposed Low-Rank Adaptation) (Liu et al., 2024b) presents a novel approach as illustrated in Figure 8 (c) by decomposing model weights $W_0 \in \mathbb{R}^{d \times k}$ into magnitude and direction as follows:

$$W_0 = m\frac{V}{\|V\|_c} = \|W_0\|_c \frac{W_0}{\|W_0\|_c}, \tag{17}$$

where $m \in \mathbb{R}^{1 \times k}$ is the magnitude vector, $V \in \mathbb{R}^{d \times k}$ is the directional matrix, with $\|\cdot\|_c$ being the vector-wise norm of a matrix across each column. Subsequently, DoRA adopts a unique fine-tuning strategy for $m$ and $V$. While both are tunable, only $V$ undergoes LoRA reparameterization, defined as:

$$W' = \underline{m}\frac{V + \underline{\Delta V}}{\|V + \underline{\Delta V}\|_c} = \underline{m}\frac{W_0 + \underline{W_{\text{up}}W_{\text{down}}}}{\|W_0 + \underline{W_{\text{up}}W_{\text{down}}}\|_c}, \tag{18}$$

where $\Delta V$ is the incremental directional update learned by LoRA, and the underlined parameters denote the trainable parameters. Through this methodology, DoRA consistently outperforms LoRA across various tasks and models, demonstrating its superiority.

### 3.4 Hybrid PEFT

The efficacy of various PEFT methods can significantly differ across different tasks. As a result, numerous studies aim to either combine the advantages of diverse PEFT approaches or seek to establish a unified perspective by analyzing the similarities among these methods. For instance, **UniPELT** (Mao et al., 2021) integrates LoRA, prefix-tuning, and adapters into each Transformer block. To control which PEFT submodules should be activated, they also introduce a gating mechanism. This mechanism consists of three small FFNs that each produce a scalar value $\mathcal{G} \in (0, 1)$, which is then applied to the LoRA, prefix, and adapter matrices, respectively. Across various setups, UniPELT has consistently shown improvements in accuracy ranging from 1% to 4%. **S4** (Chen et al., 2023a) explores design spaces for several PEFT methods (i.e., Adapter (A), Prefix (P), BitFit (B), and LoRA (L)) to uncover underlying design patterns. After a series of experiments, their findings include: (1) Applying the spindle grouping partitioning for Transformer layers, which results in four layer groups $G_i$ for $i \in \{1 \ldots 4\}$. Layers in one group have similar behaviors together, which means should apply similar PEFT strategies. (2) Allocating the number of trainable parameters to layers uniformly. (3) Tuning all the groups. (4) Assigning different PEFT strategies to different groups. The resulting design space that has the best performance is:

$$G_1 : (A, L), G_2 : (A, P), G_3 : (A, P, B), G_4 : (P, B, L)$$

**MAM Adapter**(He et al., 2021) explores the intrinsic similarity between three additive PEFT methods: adapters, prefix-tuning, and LoRA, which leads to the development of three variants: *Parallel Adapter*, which places adapter layers alongside specific layers (SA or FFN) instead of after them; *Multi-head Parallel Adapter*, which divides the parallel adapter into multiple heads, each affecting the head attention output in SA; and *Scaled Parallel Adapter*, which adds a scaling term after the parallel adapter layer, similar to LoRA. Extensive experimentation revealed that the most effective configuration involves using prefix-tuning in the SA layer and the scaled parallel adapter in the FFN layer, which is called the MAM Adapter. **LLM-Adapters** (Hu et al., 2023a) builds an easy-to-use framework that incorporates various PEFT techniques into LLMs. Through comprehensive benchmarking across multiple datasets, the study reveals several key insights: (1) The most effective locations for series adapters, parallel adapters, and LoRA are after the MLP layers, alongside the MLP layers, and simultaneously following the Attention layers and MLP layers, respectively. (2) Smaller LLMs utilizing PEFT can achieve competitive or even superior results on certain tasks when compared to their larger counterparts. (3) With appropriate in-distribution fine-tuning data, smaller models are capable of surpassing larger models in task-specific performance.

Several studies leverage neural architecture search (NAS) to find better PEFT combination approaches. For example, **NOAH** (Zhang et al., 2022b) discovers that different PEFT configurations are specifically tailored for different tasks. To address this issue, NOAH employs NAS to identify the most effective PEFT

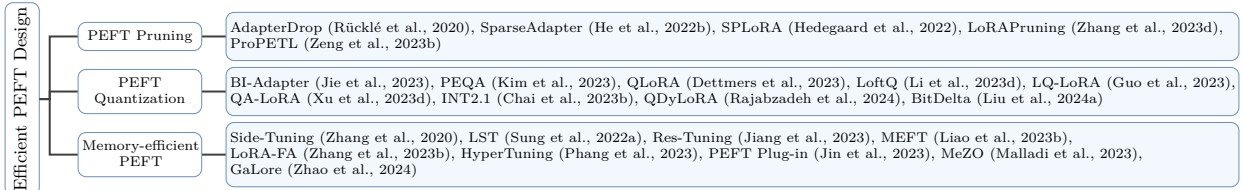

Figure 9: Taxonomy of Efficient PEFT Design.

configurations for each dataset. Specifically, NOAH's searching space encompasses three PEFT methods: Adapter, LoRA, and Visual Prompt Tuning (VPT). It utilizes AutoFormer (Chen et al., 2021a), a one-shot NAS algorithm, for the efficient discovery of optimal prompt modules. In a related vein, **AUTOPEFT** (Zhou et al., 2023) first establishes a searching space that includes serial adapters, parallel adapters, and prefix tuning. After that, they propose an effective NAS method based on a high-dimensional multi-dimensional Bayesian optimisation (Frazier, 2018). Both NOAH and AUTOPEFT demonstrate the capability of NAS in enhancing PEFT configurations across a variety of tasks.

## 4 Efficient PEFT design

Processing latency and peak memory overhead are pivotal factors to consider from a computational standpoint. This section introduces a key characteristic in LLMs aimed at balancing between latency and memory usage (Section 4.1). Following this, we explore strategies for developing efficient PEFT methods to address computational challenges, including **PEFT pruning** (Section 4.2), **PEFT quantization** (Section 4.3), and **memory-efficient PEFT techniques** (Section 4.4), each designed to enhance model performance while minimizing resource consumption. It is noteworthy that quantization inherently addresses memory overhead concerns. However, given its distinct characteristics, we address these quantization methods separately rather than incorporating them under the memory-efficient PEFT section.

### 4.1 KV-cache Management for PEFT Efficiency

The core of the LLMs model lies in an auto-regressive Transformer model. When we consider the auto-regression characteristic, it becomes a major challenge in designing an inference system, because every time a new token is generated, the entire LLM model has to transfer all the weights from different memories to the memory of the graphics processor, which is very unfriendly to single-user task scheduling or multi-user workload balance. The challenging part of serving the auto-regressive paradigm is that all previous sequences have to be cached and saved for the next proceeding iteration; the cached activation generated from the previous sequences is stored as the Key-Value Cache (KV-cache). To effectively manage these challenges, S-LoRA Sheng et al. (2023a) employs a Unified Paging mechanism within a unified memory pool that dynamically allocates and manages memory in a paged fashion. This sophisticated approach minimizes memory fragmentation and enhances the efficiency of KV-cache storage by allowing for flexible and efficient memory access patterns. These pages are managed such that the KV-cache associated with each adapter is segmented into manageable blocks, streamlining access and reducing the overhead associated with variable cache sizes. By dynamically adjusting to different KV-cache requirements, S-LoRA maintains high throughput and performance, ensuring that the system remains responsive and efficient even as it scales to serve thousands of adapters simultaneously. This efficient handling of KV-cache is crucial for supporting the auto-regressive nature of LLMs in high-demand environments, optimizing both single-user and multi-user workload balancing.

### 4.2 Pruning Strategies for PEFT

The inclusion of pruning can substantially enhance the efficiency of PEFT methods. In particular, **AdapterDrop** (Rücklé et al., 2020) explores the removal of adapters from lower transformer layers and multi-task adapters in AdapterFusion (Pfeiffer et al., 2020), which shows that the pruning can improve the training

and inference efficiency with minimal decrease in performance. **SparseAdapter** (He et al., 2022b) investigates different pruning methods and finds that high sparsity ratio (80%) can outperform standard adapters. Additionally, the *Large-Sparse* configuration, which increases the bottleneck dimension while maintaining a constant parameter budget (e.g., doubling dimensions with a 50% sparsity), substantially enhances the model's capacity, resulting in improved performance. **SPLoRA** (Hedegaard et al., 2022) adopts channel-based pruning to the LoRA weights $W_{\text{down}}$ and $W_{\text{up}}$. This pruning affects not only the source weights $W_0$, but also the LoRA parameters $W_{\text{up}}$ and $W_{\text{down}}$. Similarly, **LoRAPruning** (Zhang et al., 2023d) adopts structured pruning not only to the pre-trained model weights but also to the LoRA weights. In contrast to unstructured LoRA pruning methods, which primarily focus on sparsifying model weights while leaving LoRA weights dense, thus making weight merging challenging to achieve, LoRAPruning enables the weights to be merged easily. Additionally, this work also introduces a novel criterion that utilizes LoRA's gradients as an approximation of the gradients for the pre-trained weights, enabling the estimation of weight importance. **ProPETL** (Zeng et al., 2023b) constructs a single shared *prototype* (e.g., adapter, prefix, or LoRA) across layers and tasks. In addition, ProPETL learns binary masks to prune different sub-networks in different layers and tasks. As a result, the parameters can be reused across layers and tasks, largely increasing the parameter efficiency.

## 4.3   Quantization Strategies for PEFT

Quantization serves as another popular technique for improving computational efficiency and reducing memory usage. For example, by investigating the loss landscape of adapters, **BI-Adapter** (Jie et al., 2023) finds that adapters are resistant to noise in parameter space. Building on this insight, the authors introduce a clustering-based quantization approach. Remarkably, they demonstrate that a 1-bit quantization of adapters not only minimizes storage requirements but also achieves superior performance among all precision settings. **PEQA** (Parameter-Efficient and Quantization-aware Adaptation) (Kim et al., 2023) uses a two-stage pipeline to achieve parameter-efficient and quantization-aware fine-tuning. In the first stage, the pre-trained FFN weight matrix $W \in \mathbb{R}^{n \times m}$ is quantized to $W = s \cdot \overline{W}$, where $s \in \mathbb{R}^{n \times 1}$ represents per-channel scales and $\overline{W}$ denotes the quantized weight. In the second stage, $\overline{W}$ remains fixed, and fine-tuning is only conducted on $s$. This approach not only ensures memory efficiency but also facilitates parameter efficiency. **QLoRA** (Dettmers et al., 2023) proposes several novel techniques, including a 4-bit NormalFloat, a Double Quantization, and a Paged Optimizers, to backpropagate a 4-bit quantized pretrained language model into LoRA. These techniques enable the fine-tuning for a 65B language model on a single 48GB GPU while maintaining similar performance to the full 16-bit fine-tuning. Similar to the original implementation (Hu et al., 2021), QLoRA attaches the fixed zero-initialized LoRA weights to the quantized pre-trained model as the training start point. However, when applying the extreme low-bit (e.g., 2-bit) quantization, the huge quantization error can adversely impact the initialization of LoRA fine-tuning, i.e., $quantization(W_0) + W_{\text{down}} W_{\text{up}} \neq W_0$ where $W_{\text{down}} = \mathbf{0}$, which will harm the fine-tuning performance as shown in the work by Liao et al. (2023b). To solve this, several quantization strategies are proposed to eliminate the quantization error. For example, **LoftQ** (LoRA-Fine-Tuning-aware Quantization) (Li et al., 2023d) presents an innovative framework that provides a superior initialization point of quantized backbone weights and LoRA weights for subsequent LoRA fine-tuning. This approach addresses the discrepancies caused by quantization through the optimization of a Frobenius norm objective during network initialization, which takes both the LoRA weights and the quantized pre-trained backbone into consideration. LoftQ exhibits superior performance in 2-bit quantization over QLoRA, as well as greater generalization for downstream tasks. **LQ-LoRA** (Guo et al., 2023) uses an iterative algorithm inspired by robust principal components analysis (Zhou & Tao, 2011; Wright et al., 2009) which decomposes the weight $W_0$ such that $W_0 \approx Q + L_1 L_2$ to resolve the inaccuracy caused by the quantization error, where $Q$ is the quantized component which remains fixed and $L_1 L_2$ is the trainable low-rank component. Moreover, this approach leverages integer linear programming to determine a mixed quantization strategy, enabling dynamic quantization configurations for each weight matrix while adhering to a predetermined total bit rate limit. **QA-LoRA** (Xu et al., 2023d) address another limitation of QLoRA, which struggles to preserve its quantized property post-fine-tuning. In QLoRA, the quantized pre-trained weight (NF4) has to be recovered to FP16 to match the LoRA weight precision (FP16) during weight merging. Instead, QA-LoRA uses INT4 quantization and introduces group-wise operators to enable quantization during the inference stage, therefore improving the efficiency and accuracy

compared with QLoRA. **BitDelta** (Liu et al., 2024a) introduces a novel 1-bit post-training quantization method that acts on the weight delta between a fine-tuned model and its underlying pre-trained model. Specifically, given the weight matrices $W_{\text{fine}}$ and $W_{\text{base}}$ from the fine-tuned and base models respectively, the weight delta $\Delta = W_{\text{fine}} - W_{\text{base}}$ is binarized as $\hat{\Delta} = \alpha \odot \text{Sign}(\Delta)$. Here, $\alpha$, a high-precision scalar, is initialized based on the mean absolute delta value $\alpha = \frac{1}{nm} \sum_{ij} |W_{ij}|$, with $\text{Sign}(\cdot)$ indicating the sign of $\Delta$. BitDelta further calibrates the scaling factors via distillation on a compact calibration dataset, while the binary matrices remain unchanged. This approach notably streamlines the deployment of multiple fine-tuned models on shared servers by utilizing a singular full-precision base model alongside efficiently batched 1-bit deltas.

## 4.4 Memory-efficient PEFT Methods

Fine-tuning the full LLMs necessitates substantial training memory owing to their considerable size. While most PEFT methods primarily target parameter efficiency, they still incur a significant memory overhead during training because gradient computation and backpropagation are still necessary for these methods. For example, prevalent PEFT techniques such as adapters and LoRA can only reduce memory usage to approximately 70% compared to full model fine-tuning according to some literature (Sung et al., 2022a; Jin et al., 2023). From a computational perspective, memory efficiency also remains a critical factor that cannot be overlooked.

To improve memory efficiency, various techniques have been developed to minimize the need for caching gradients for the entire LLM during fine-tuning, thereby reducing memory usage. For example, both **Side-Tuning** (Zhang et al., 2020) and **LST** (Ladder-Side Tuning) (Sung et al., 2022a) introduce a learnable network branch parallel to the backbone model. By channeling the backpropagation exclusively through this parallel branch, it circumvents the need to store gradient information for the main model's weights, thus markedly reducing memory requirements during training. Similarly, **Res-Tuning** (Jiang et al., 2023) disentangles the PEFT tuners (e.g., prompt tuning, adapter) from the backbone model. On top of the disentanglement, a memory-efficient fine-tuning framework named Res-Tuning-Bypass is proposed, which generates a bypass network in parallel with the backbone model by removing the data flow from the decoupled tuners to the backbone. This eliminates the requirement for gradient caching within the backbone model during backpropagation. **MEFT** (Liao et al., 2023b) (memory-efficient fine-tuning) is an approach inspired by the *reversible model* (Gomez et al., 2017). During the training of a reversible model, intermediate activations are not required to be cached in the forward pass. During backpropagation, they can be recalculated from the final output. To save the memory during fine-tuning, MEFT investigates how to transform an LLM to its reversible counterparts without additional pre-training. A critical aspect of this transformation is the careful initialization of newly introduced parameters in the pre-trained models. MEFT demonstrates the importance of parameter initialization and suggests that these parameters must be initialized in a manner that preserves the pre-trained model's starting point, ensuring that the fine-tuning of the modified model achieves performance on par with full fine-tuning methods. With this key consideration, MEFT introduces three distinct methods, each significantly curtailing the memory demands traditionally required for storing activations. **LoRA-FA** (Zhang et al., 2023b) addresses a limitation about memory overhead in LoRA fine-tuning. During training, LoRA modules still require high activation memory consumption. This is because, during backpropagation, large input activations must be stored during the forward pass to compute gradients. LoRA-FA resolves this issue by freezing both the pre-trained weights $W_0$ and the projection-down weights $W_{\text{down}}$, and only updating the projection-up weights $W_{\text{up}}$. Consequently, the input activation $h_{in}$ no longer needs to be stored, as the intermediate activation $W_{\text{down}} h_{in}$ is adequate for gradient computation for $W_{\text{up}}$. Given that $r \ll d$, the memory requirement for activations in LoRA-FA can be significantly reduced.

To further reduce memory usage during fine-tuning, some methods attempt to circumvent backpropagation within LLMs to address this issue. **HyperTuning** (Phang et al., 2023) employs a HyperModel to generate PEFT parameters using only fewshot examples. This approach demonstrates results comparable to those obtained through full model fine-tuning. **PEFT Plug-in** (Jin et al., 2023) first trains PEFT modules on small language models, which is more memory efficient compared to training on large ones. Subsequently, the research introduces a suite of techniques for seamlessly integrating these trained PEFT modules into LLMs during inference. This strategy effectively circumvents the necessity of gradient-based optimization

directly on the larger models, resulting in substantial memory savings. However, it is important to note that both HyperModel and PEFT Plug-in still require additional model training, and this training cost cannot be entirely overlooked. **MeZO** (Malladi et al., 2023) introduces a memory-efficient zeroth-order (ZO) optimizer for LLMs. Unlike conventional PEFT techniques, which rely on backpropagation to compute gradients for updating model parameters, MeZO fine-tunes LLMs through only forward passes. It accomplishes this by employing a ZO gradient estimator to calculate the gradient. Notably, MeZO implements an in-place solution for the classic ZO gradient estimator, effectively mitigating memory consumption during inference execution. This innovative approach allows for efficient fine-tuning of LLMs containing 30 billion parameters on a single GPU with 80GB of memory, all while maintaining performance that is comparable to fine-tuning using backpropagation. Furthermore, it can substantially decrease storage demands in comparison to the traditional PEFT methods such as LoRA and adapters.

# 5 PEFT for DNNs of Other Applications

In Section 3, we outlined four categories of PEFT methods along with their improvements. Nonetheless, our discussion did not fully extend to the utilization or adaptation of PEFT techniques beyond traditional architectures (e.g., LLMs) or standard benchmarks (e.g., the GLUE dataset), where the majority of the discussed PEFT methods are applied. Therefore, in this section, we will highlight and discuss several most representative works that leverage PEFT strategies for various downstream tasks. In this section, we do not aim to cover all PEFT application scenarios. Our objective is to showcase the significant influence of PEFT within various research domains and demonstrate how to optimize and tailor general-purpose PEFT methods to achieve enhanced performance in specific models or tasks.

Typically, fine-tuning happens when adapting a pre-trained backbone model to specialized downstream tasks. To this end, this section organizes the discussion around various model architectures, which include: LLM, Vision Transformer (ViT), Vision-Language Alignment Model (VLA), and Diffusion model. Within each architectural category, the discussion is further classified based on different downstream tasks.

## 5.1 PEFT for LLMs – Beyond the Basics

Instead of common tasks in NLP such as NLU and NLG, PEFT techniques boast a wide array of applications across diverse scenarios. PEFT has been successfully implemented in commonsense question answering (Huang et al., 2023c; Zhao et al., 2023d), multi-level implicit discourse relation recognition (Zhao et al., 2023c), out-of-distribution detection (Ouyang et al., 2023), privacy protection (Ozdayi et al., 2023; Xiao et al., 2023b), federated learning (Che et al., 2023), and social biases mitigation (Li et al., 2023c). In this section, we pay more focus on three representative downstream tasks: visual instruction following, continual learning, and context window extension.

### 5.1.1 Visual Instruct Following

Several studies, including VL-BART (Cho et al., 2021), MiniGPT-4 (Zhu et al., 2023b), and LLaVA (Liu et al., 2023a), have successfully extended the capabilities of LLMs, initially designed for pure text, to comprehend and generate responses to visual inputs. These enhanced models, namely visual instruct-following LLMs, can process both images and text to produce textual responses, which can be benchmarked on tasks such as image captioning (Rennie et al., 2017; You et al., 2016; Vinyals et al., 2016; Hossain et al., 2019) and visual question answering (VQA) (Wang et al., 2017; Wu et al., 2017; Antol et al., 2015). However, these methods fine-tune the entire LLM to learn the visual representations, which can be inefficient in both time and memory. Therefore, it is natural to apply PEFT techniques in the fine-tuning of visual instruct-following LLMs. An earlier work **VL-Adapter** (Sung et al., 2022b) directly applies several PEFT methods (Adapter (Houlsby et al., 2019), Hyperformer (Mahabadi et al., 2021) and Compacter (Karimi Mahabadi et al., 2021)) on VL-BART (Cho et al., 2021) then benchmarks them on several image-text and video-text tasks. Results show that vanilla adapters are the best among them, which can achieve performance on par with full fine-tuning. However, considering the functionality gap between the encoders and decoders in VL-BART, directly assigning identical modular modifications will lead to suboptimal performance. Therefore,

**VL-PET** (Hu et al., 2023b) selectively integrates PEFT modules into different components of the encoder and decoder. They also introduce a granularity-controlled mechanism for finer-grained control.

To adapt the recently prevalent LLaMA model, **LLaMA-Adapter** (Zhang et al., 2023f) prepends a set of learnable prompts (similar to prefix tuning) to the input tokens in LLaMA's higher transformer layers. To avoid the unstable fine-tuning with large loss values at early training stages, instead of the randomly initialized weights of other PEFT methods, LLaMA-Adapter adopts a zero-initialized attention mechanism, which learns a zero-initialized gating factor to adaptively control the contribution of adaptation prompts to the word tokens. This can maintain the fine-tuning starting point the same as the original model and progressively inject new knowledge into the model, where a similar idea can be found in MEFT (Liao et al., 2023b) and LoftQ (Li et al., 2023d) discussed earlier. To represent visual information, LLaMA-Adapter extracts multi-scale global image features using a CLIP image encoder and then projects them to linguistic embedding space. After that, the feature is element-wisely added onto the adaptation prompts at all inserted transformer layers. LLaMA-Adapter only introduces 1.2M learnable parameters in LLaMA-7B and costs less than one hour for fine-tuning on 8 A100 GPUs. A following work **LLaMA-Adapter V2** (Gao et al., 2023b) demonstrates that the simple multimodal fusion in LLaMA-Adapter cannot generalize to more challenging open-ended multimodal reasoning tasks, where the visual cues tend to dominate the adaptation prompts than the language instruction data. To address this, LLaMA-Adapter V2 decouples the learning of instruction-following ability (to generate long language responses) and vision-language alignment to avoid interference between visual and language fine-tuning. Specifically, LLaMA-Adapter V2 sets disjoint parameter groups which are respectively learned from image-text pairs and language instruction data. The visual adaptation prompts are inserted in the early stage of LLM, while the language adaptation prompts remain at the higher transformer layers similar to the LLaMA-Adapter. Additionally, LLaMA-Adapter V2 introduces more learnable parameters and several expert systems (e.g., captioning, detection, and OCR) to enhance multimodal performance. **LayerNorm Tuning** (Zhao et al., 2023a) adjust only the weights of the LayerNorm within each attention block. This straightforward technique can achieve comparable or even better performance than the finetuning, while offering about $10\times$ more parameter efficiency than LoRA.

### 5.1.2 Continual Learning

Continual Learning (CL) aims to learn a sequence of new tasks over time within one single model, which has broad application in scenarios such as dialogue systems (Lee, 2017), information extraction systems (Chang et al., 2006), and question answering systems (Yang et al., 2019). The main challenge in CL is catastrophic forgetting (Kirkpatrick et al., 2017). A popular practice, called architecture-based methods, tackles the CL by maintaining task-specific parameters in the model for each new task. Therefore, it's natural to leverage PEFT methods for CL tasks (Madotto et al., 2020; Zhu et al., 2022; Dai et al., 2022; Liang et al., 2023). For example, **AdapterCL** (Madotto et al., 2020) parameterizes each new task using residual adapters. During testing, since the task-id is not provided, AdapterCL uses an entropy-based classifier to select which adapter to use for accomplishing a specific task. **CPT** (Continual Prompt Tuning) (Zhu et al., 2022) trains a soft prompt for each task. Instead of training soft prompts from scratch, CPT proposes a series of techniques (continual prompt initialization, query fusion, memory replay, and a memory-guided technique) to achieve knowledge transfer from preceding and subsequent tasks. **O-LoRA** (orthogonal low-rank adaptation) (Wang et al., 2023b) employs a strategy of learning distinct tasks within separate low-rank vector subspaces that are kept orthogonal to each other in order to minimize interference. This approach can effectively reduce catastrophic forgetting during the acquisition of new tasks.

### 5.1.3 Context Window Extension

LLMs are typically trained with a pre-defined context size. For example, LLaMA and LLaMA2 have pre-defined context sizes of 2048 and 4096 tokens, respectively. The positional encoding RoPE has weak extrapolation properties (Chen et al., 2023d), which means the performance drops obviously given an input length exceeds the pre-defined context length. To solve this, a naive solution is to fine-tune a pre-trained LLM to a longer context. However, this escalates computational costs quadratically with context size, straining memory and processing resources. To address this, **LongLoRA** (Chen et al., 2023e) proposes to fine-tune a pre-trained LLM using LoRA to enlarge the context size. To reduce the perplexity gap between LoRA

tuning and full fine-tuning, LongLoRA also opens embedding and normalization layers for training. In order to further improve training efficiency in a long context scenario, LongLoRA further introduces a novel shifted sparse attention ($S^2$-Attn) as an efficient substitute for standard self-attention during training. A subsequent study **LongQLoRA** (Yang, 2023) combines the advantages of LongLoRA with QLoRA and Position Interpolation (Su et al., 2021a) to save GPU memory. This work successfully extends the context length of LLaMA2-13B from 4096 to 8192 on a single V100 with 32GB memory. **LLoCO** (Tan et al., 2024) introduces a pipeline that learns contexts offline through the combination of context compression and LoRA. The process begins by compressing documents into compact contexts, then fine-tuning LLM using LoRA on the compacted context to improve the LLM's ability to accurately extract and utilize information from these compressed representations. During model serving, a standard RAG retriever selects both the compressed document and the most relevant LoRA module, and applies them to the LLM for inference. This approach effectively extends the context window of a 4k token LLaMA2-7B model to handle up to 128k tokens.

In addition to limited training-stage sequence length, real-world system memory constraints introduce another critical bottleneck to the context window. Specifically, the capacity of the KV-cache is curtailed by available system memory. For example, a 30B parameter LLM operating with an input length of 1024 and a batch size of 128 might necessitate up to 180GB for the KV-cache (Zhang et al., 2024b), thereby restricting the feasible size of the context window. In response to this, some strategies have resorted to quantizing the KV cache (Sheng et al., 2023b; Dettmers et al., 2022), but quantization will certainly compromise performance. To effectively counteract this issue without significant loss, **GEAR** (Kang et al., 2024) presents a novel approach by employing a low-rank matrix to capture the majority of coherent bases of quantization error, complemented by a sparse matrix that addresses errors from outlier entries, thus efficiently minimizing approximation errors.

## 5.2 PEFT for ViTs

ViT (Dosovitskiy et al., 2010) has emerged as a powerful backbone model in the recent computer vision community. In the ViT model, images are treated as sequences of fixed-size patches analogous to how LLM uses discrete tokens. These patches undergo linear embedding and then receive positional encodings. Subsequently, they are processed through standard Transformer encoders. The training of ViT can be supervised (Dosovitskiy et al., 2010; Steiner et al., 2021) or self-supervised (Chen et al., 2021b; He et al., 2022a), and ViT can achieve superior performance when training with more data and using larger model size (Dehghani et al., 2023). However, such scaling up inevitably escalates training and storage costs. Therefore, similar to LLMs, PEFT is widely implemented in various downstream tasks, such as dense prediction (Chen et al., 2022b), continual learning (Wang et al., 2022b; Gao et al., 2023c), deep metric learning (Ren et al., 2024). Here, we focus on two typical tasks to showcase the involvement of PEFT: image classification and video recognition.

### 5.2.1 Image Classification

Image classification on targeted visual datasets is a very common demand and has extensive applications, while pre-train then fine-tuning paradigm serves as a widespread strategy. A variety of methods leverage PEFT techniques to achieve efficient model tuning (Jia et al., 2022; Chen et al., 2022b;a; Jie & Deng, 2022). For instance, **AdaptFormer** (Chen et al., 2022a) inserts adapter modules in parallel to the FFN of the original ViT model for visual recognition tasks. **VPT** (Visual Prompt Tuning) (Jia et al., 2022) prepends a small amount of task-specific parameters into the input sequence of each Transformer layer. When applying ViT to downstream tasks, only these added parameters and the classification head are set to trainable. **Yoo et al. (2023)** notices that compared with supervised ViT, VPT often underperforms with self-supervised ViT. Further analysis demonstrates that different pre-trained methods and downstream tasks have varying degrees of dependency on transformer blocks at different locations. To tackle this issue, the research introduces adaptable gates for ViT blocks. These gates dynamically modulate the contribution of prompt tokens to ViT blocks, allowing for a more targeted adaptation of the model to the task at hand.

### 5.2.2 Video Recognition

Several works consider the more challenging adaptation problem that transfers ViT to downstream tasks that have a much larger domain gap. For example, **ST-Adapter** (Spatio-Temporal Adapter) (Pan et al., 2022) and **AIM** (Yang et al., 2023c) both insert adapters layers into pre-trained ViT blocks. Their primary goal is to model spatial-temporal information, thereby enabling efficient adaptation of ViTs from image models to video tasks. Notably, both methodologies have exhibited performance that surpasses traditional full-model fine-tuning approaches.

## 5.3 PEFT for VLAs

Vision-language alignment models (VLA), such as CLIP (Radford et al., 2021), ALIGN (Jia et al., 2021), DeCLIP (Li et al., 2021), and FLAVA (Singh et al., 2022), are designed to learn a good image and text features which can be aligned within a unified representation space. Each VLA typically consists of separate image and text encoders that extract respective features. Contrastive learning is leveraged in these models to effectively align the image and text features. Fine-tuning is leveraged to improve the performance of VLA in specific datasets or tasks, but fine-tuning the full model is computationally intensive. For instance, fine-tuning CLIP RN50x64 requires a batch size of 32,768 and 18 days of training on 592 V100 GPUs (Radford et al., 2021). Moreover, full fine-tuning on smaller datasets often leads to catastrophic forgetting (Kirkpatrick et al., 2017). In response to these challenges, and drawing inspiration from the success of PEFT techniques in NLP, a range of PEFT strategies have been proposed and implemented in VLA models, such as semantic segmentation (Xu et al., 2023c; Yu et al., 2023; Xu et al., 2023e), point cloud understanding (Zhang et al., 2022a; Zhu et al., 2023d; Wang et al., 2022c; Huang et al., 2023b), video understanding (Ju et al., 2022; Ni et al., 2022; Lin et al., 2022), visual reasoning (Han et al., 2023b; Doveh et al., 2023), temporal action detection (Nag et al., 2022), to name a few. This section will focus on one common task that uses VLAs: open-vocabulary image classification.

### 5.3.1 Open-vocabulary Image Classification

In open-vocabulary image classification, earlier works design class-specific prompts, e.g., *a photo of a [CLASS]*, for each category, and rank images based on their similarity to these textual descriptions. **CoOp** (Context Optimization) (Zhou et al., 2022b) replaces the handcrafted text prompt with learnable vectors, while keeping the entire VLA fixes during training. **CoCoOp** (Conditional Context Optimization) (Zhou et al., 2022a) builds on this by tackling CoOp's limitations in generalizing to unseen classes. It introduces a lightweight neural network that generates an input-specific context token, dynamically adapting the prompt based on each image, thereby enhancing generalizability, but at the cost of increased computational demands due to the instance-aware operation. **ProGrad** (Zhu et al., 2023a) addresses the over-fitting risk in CoOp in a few-shot setting by regularizing the soft prompt updates whose gradient is aligned to the general knowledge only updates the prompt whose gradient is aligned (or non-conflicting) to the general knowledge offered by the original prompt. **MaPLe** (Khattak et al., 2023) notes that existing methods learn prompts either in the language or in the vision branch of CLIP, which is not efficient in leveraging the multimodal nature of VLAs. To address this, MaPLe proposes branch-aware hierarchical prompts that simultaneously adapt both language and vision branches, and achieves superior performance. **TPT** (test-time prompt tuning) (Shu et al., 2022) studies prompt tuning on the fly without additional training samples. Specifically, during inference, TPT first augments the input image into various views, which are then utilized to tune the learnable prompts. The primary training objective is to ensure the VLA can generate consistent responses when faced with these differing views. A following work **DiffTPT** (Feng et al., 2023) further enhances the data diversity of test samples through diffusion models.

In another direction, several studies explore the usage of adapters in VLA. For example, **CLIP-Adapter** (Gao et al., 2023a) integrates residual-style adapters after CLIP's text and visual encoders. Therefore, unlike CoOp and CoCoOp, CLIP-Adapter avoids the gradient backpropagation through CLIP's encoders, leading to reduced computational requirements in terms of both training memory and time. **Tip-Adapter** (Zhang et al., 2021) adopts the same design with CLIP-Adapter. Different from CLIP-Adapter, the weights of the adapter are obtained in a training-free manner from a query-key cache model (Orhan,

2018; Grave et al., 2017) constructed from few-shot supervisions in a non-parametric manner. As a result, Tip-Adapter exhibits great efficiency compared to CLIP-Adapter's SGD training process.

## 5.4 PEFT for Diffusion Models

Diffusion models (Ho et al., 2020; Sohl-Dickstein et al., 2015) are a class of generative models that learn to generate data by transforming random noise into a structured output by a progressive denoising process. During training, diffusion models learn to reverse the noise added to training data using a denoising network, while in inference, they start from noise, using a denoising network to iteratively create data that mirrors the same distribution as the training examples. Diffusion models have various applications (Han et al., 2023a; Yang et al., 2023b; Croitoru et al., 2023; Dhariwal & Nichol, 2021; Ruiz et al., 2023), while the most notable is stable diffusion (Rombach et al., 2022), which bridges the gap between text and image with its robust capability to generate coherent and contextually relevant images directly from textual descriptions. Numerous studies leverage PEFT techniques to adapt a pre-trained diffusion model for downstream tasks, including accelerating sampling speed (Luo et al., 2023; Chai et al., 2023a), text-to-video adaptation (Wu et al., 2023a; Xing et al., 2023), text-to-3D adaptation (Zeng et al., 2023a), etc. This section mainly focuses on two scenarios: integrating additional input modalities beyond mere text-based conditioning, and customizing content generation based on pre-trained diffusion model.

### 5.4.1 Additional Input Control

To incorporate additional input modalities (e.g., layout, keypoints) while retaining the extensive knowledge in the pre-trained model, **GLIGEN** introduces a novel approach, which maintains the original model's weights intact and integrates new, trainable gated Transformer layers (Alayrac et al., 2022) that take in the new grounding input. The resulting model can not only accurately represent the grounding conditions but also produce high-quality images. Remarkably, the model can also generalize well to unseen objects during inference. **ControlNet** (Zhang et al., 2023c) fine-tunes a trainable copy of the encoding layers from Stable Diffusion while locking its pre-trained parameter weights. The fixed original model and the trainable copy are bridged through zero convolution layers. These layers, starting with zero-initialized weights, are designed to progressively adapt during training, ensuring that harmful noise does not affect the pre-trained features of Stable Diffusion at the beginning of training. This refined model is capable of conditioning on a variety of inputs such as Canny edges, Hough lines, user scribbles, human key points, segmentation maps, shape normals, depths, etc. **Concept Sliders** (Gandikota et al., 2023) introduces a plug-and-play LoRA adaptors to allow precise editing of concepts (e.g., age, smiling) within a diffusion model. **T2I-Adapter** (Mou et al., 2023) introduces a lightweight adapter model designed to align external control signals with the internal knowledge of text-to-image diffusion models. This adapter enables precise manipulation through structural control (e.g., sketch, depth map, semantic segmentation map, and keypose), color control (e.g., hue and color distribution), and integrating various controls by composing multiple adapters.

### 5.4.2 Customized Generation

The effectiveness of text-to-image diffusion models is limited by the user's ability to articulate the desired target through text descriptions. For instance, it is difficult to describe the precise features of an innovative toy car which is not encountered during large-scale model training. Consequently, the objective of customized generation is to enable the model to grasp new concepts from a minimal set of user-supplied images. **Textual Inversion** (Gal et al., 2022) addresses this by finding a new pseudo-word $S_*$ (similar to soft prompt discussed in Section 3.1.2) that represents new, specific concepts in the textual embedding space of pre-trained text-to-image diffusion models. The pseudo-word $S_*$ is optimized via the original optimization goal in diffusion models given a small image set (typically 3-5 images) depicting the concept, and the pre-trained model is left untouched. During inference, $S_*$ can be treated like any other word and composed with other textual queries (e.g., "a photo of $S_*$ on the beach"). **Custom Diffusion** (Kumari et al., 2023) tackles a more challenging setting: compositional fine-tuning of multiple concepts. It fine-tunes only the $W_k, W_v$ mapping from text to latent features in attention layers, which yields superior performance in multi-concept learning scenarios. Additionally, during fine-tuning, Custom Diffusion prevents model forgetting by introducing a small set of real images with captions akin to the target, alongside employing augmentation for faster

convergence and improved results. **IP-Adapter** (Ye et al., 2023) identifies limitations in current approaches (e.g., ControlNet and T2I-Adapter) which project condition signals into the cross-attention modules. When handling image conditions aiming at controlling content, these methods are unable to generate images faithful to the prompted image. The issue stems from that merging image features and text features within cross-attention layers loses image-specific information, leading to only coarse-grained controllable generation such as image style rather than image content. To overcome this, IP-Adapter introduces a novel decoupled cross-attention mechanism to distinguish between text and image features. IP-Adapter adds an additional cross-attention layer exclusively for image features in each cross-attention layer, and only the parameters of the new cross-attention layers are trained.

## 6 System Design Challenge for PEFT

### 6.1 System design for PEFT

In this section, we begin by providing a concise overview of cloud-based PEFT systems and analyzing the design challenges. These include the efficient handling of numerous task-specific queries via centralized PEFT query servicing, the resolution of privacy and data transmission issues through distributed PEFT training, and the complexities associated with concurrent multi-PEFT training processes. Centralized systems are required to process a substantial volume of queries with minimal latency and maximal throughput. Distributed training frameworks must address privacy concerns and the computational inefficiencies that arise from data exchanges between users and cloud services. Furthermore, multi-PEFT training necessitates the optimization of memory utilization, the management of simultaneous model training, and the formulation of system architectures capable of supporting multi-tenant workloads effectively. These challenges underscore the imperative for innovative approaches to improve scalability, safeguard privacy, and optimize resource allocation in PEFT system architectures. Following this, we present the corresponding metrics employed for evaluating the system performance. Furthermore, we delve into three prospective utilization scenarios to illustrate the challenges in system design.

#### 6.1.1 Centralized PEFT Query Serving

Cloud providers have recently introduced a range of LLM services aimed at providing user applications through application programming interfaces (APIs) (OpenAI, 2023b; Team et al., 2023). These APIs facilitate the seamless integration of many machine-learning functionalities into applications. When receiving one query for one specific downstream task through API, the cloud-based server processes the query with one featured LLM model. Under this scenario, the importance of PEFT becomes apparent. Cloud providers store only a single copy of the LLM and multiple PEFT modules featuring different downstream tasks. This setup allows the LLM to maintain various branches of PEFT modules, each linked to specific API queries, i.e., PEFT queries.

Centralized PEFT query serving solutions address scenarios where multiple PEFT queries arrive in quick succession. A case study of one state-of-the-art system for this purpose is discussed in Section 6.2. Figure 10 (b) illustrates the computation pattern for multi-query PEFT inference, wherein packed PEFT queries are scheduled and executed according to their deadlines and current system conditions.

#### 6.1.2 Distributed PEFT Training

In most cases, personalized tasks are not fully supported with pre-trained models, consequently, extra fine-tuning is required to be executed with the methodologies mentioned in the previous sections. However, significant concerns arise when considering the transfer of datasets to cloud providers, given the issues related to data privacy, copyright, proprietary information, and the complexities and inefficiencies involved in data transmission. Section 6.3 gives two approaches that address this concern.

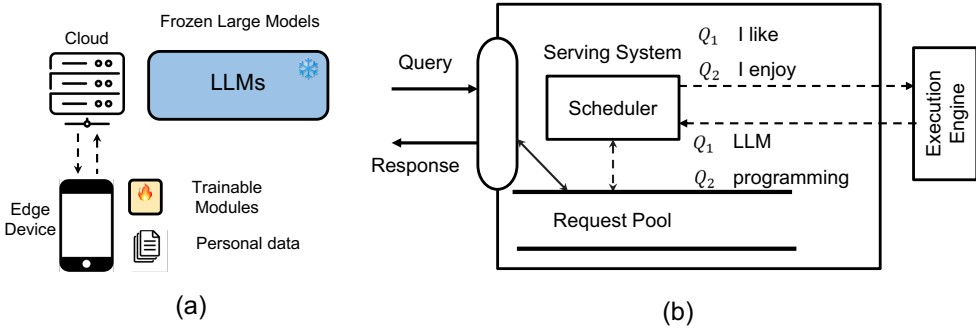

Figure 10: (a) Distributed-based system computation pattern; (b) centralized PEFT Query inference.

### 6.1.3 Multi-PEFT Training

Different from multiple-PEFT serving, tuning with multiple customized PEFTs always involves different backbone LLMs. Therefore, simultaneously tuning multiple PEFTs can pose considerable challenges. Challenges like how to manage memory gradient and model weights storage, and how to design an efficient kernel for batching PEFT training remain unsolved. PEFTs will be categorized based on their PEFT algorithms and backbone LLM models. The design challenge involves how to consolidate multiple PEFTs with the same LLM backbone and multiple different LLM backbones simultaneously. We present case studies related to this topic in Section 6.4.

### 6.1.4 Evaluation Metrics

For the proposed evaluation metrics, without loss of generality, we adopt large language models as the basis for our metric definitions.

To evaluate the system performance of *PEFT serving* systems, we propose a set of evaluation metrics:

- **System throughput**: Considering PEFT queries as inter and intra tasks, we use tokens per second to measure the system throughput.

- **Memory footprint**: Run-time memory consumption during query serving, the memory utilization comes from both model parameters and KV-cache as mentioned in Section 4.1.

- **Accuracy performance**: Real-world queries normally have different context lengths, and performance with variation length serves as a performance benchmark.

- **Quality of services**: Queries are associated with latency requirements and deadline missing rates are considered as another benchmark.

To assess the efficacy of *PEFT training* systems, we also establish a set of evaluative metrics:

- **Accuracy performance**: Performance of the fine-tuned model over the downstream tasks.

- **Compute cost**: The compute cost during forward and backward propagation operations on cloud servers and edge devices.

- **Communication cost**: Refers to the volume of data involved during the transfer of intermediate data between the edge device and the cloud.

## 6.2 Centralized PEFT Serving Frameworks

The PEFT algorithm is notable for its ability to distinguish between modifiable and immutable weights within a model. This characteristic inspires developers to amalgamate diverse LLMs with distinct PEFT

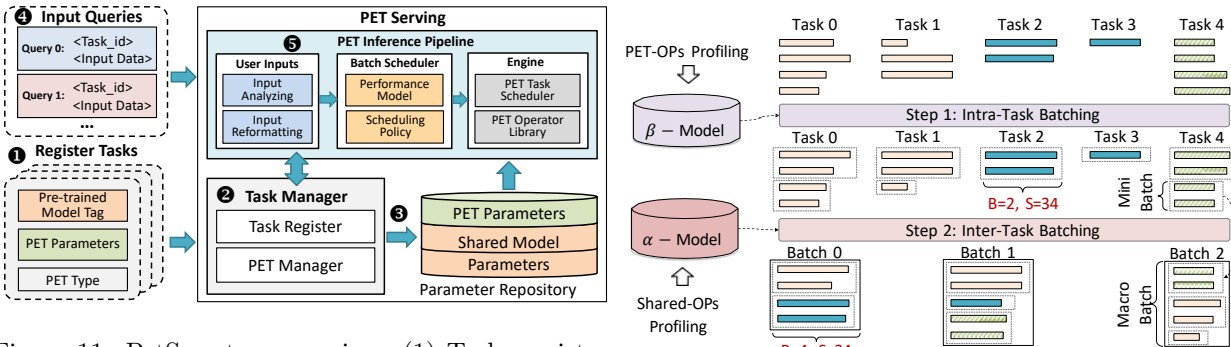

Figure 11: PetS system overview: (1) Tasks register; (2) Task manager (3) Task schedule; (4) Task serving. (Image is taken from PetS (Zhou et al., 2022c))

Figure 12: Coordinated Batching (CB) Strategy

techniques into collective units. **PetS**, as introduced in Zhou et al. (2022c), advocates for a comprehensive approach to managing multiple PEFT tasks by suggesting a unified serving framework. The framework's core advancement lies in the translation of varying PEFT tasks into integrated computation kernels to enhance efficiency. Moreover, PetS pioneers an orchestrated batching approach and a scheduling methodology, aiming to augment system throughput and leverage task parallelism respectively.

As depicted in Figure 11, the PetS framework begins with users registering PEFT tasks through a standardized Application Programming Interface (API). Upon registration, developers are expected to provide the Pre-Trained Model Tag (e.g., LLaMA), PEFT parameters in a compressed format, and the specific PEFT algorithms (e.g., LoRA, Adapter, Bitfit, etc.). These tasks are then endowed with unique identifiers, and the inference engine takes charge of query processing. PetS bifurcates the primary computational workload (e.g., linear layer computations) into three distinct computational operations: (1) Dense Matrix-Vector Multiplication (MVM) leveraging universally accessible, pre-trained weights. (2) Bias vector addition (Vadd), using either common or task-exclusive biases. (3) A combination of Sparse/dense MVM operations employing task-specific PET parameters. A unified pre-trained weight matrix $W$ is employed across PetS, facilitating the batching of initial operations, $X_t \times W$. However, subsequent task-specific computations involving PET parameters, despite being relatively minimal in complexity, are processed individually.

Considering the Adapter and Bitfit tasks as an illustration, both aim at the MLP component of LLMs. The Adapter task integrates additional weight segments, whereas Bitfit adjusts bias elements. The Adapter operation is modeled as $Y = X_{in1} \times (W + W_{ad}) + b_0$, where $X_{in1}$ represents the input for the Adapter task, $W$ and $W_{ad}$ are the original and adapter-specific PEFT weights respectively, and $b_0$ is the initial bias. The Bitfit operation, on the other hand, is defined as $Y = X_{in2} \times W + b_1$, with $b_1$ symbolizing the Bitfit-adjustable bias. These operations are further synthesized as $\{Y_1, Y_2\} = \{X_{in1}, X_{in2}\} \times W + \{X_{in1} \times W_{ad}, 0\} + \{b_0, b_1\}$, delineating that the $\{X_{in1}, X_{in2}\} \times W$ part is amenable to batching through MVM, while the $\{b_0, b_1\}$ segment pertains to the Vadd operation.

For tasks like Diff-Pruning 3.2, is a little bit different than Bitfit and Adapter. For Diff-Pruning, the computation concerning the shared weight and 'difference' are conducted separately. Then the results are added up, namely

$$X_t \times (W + \delta_t) = X_t \times W + X_t \times \delta_t$$

, here, the $W$ denotes the backbone model weights while $\delta_t$ denotes the pruned weights which can be represented as Sparse MVM.

The other challenge PetS proposed is how to schedule different PEFT requests to achieve high performance. PetS scheduler achieves high parallelism through a two-level scheduling policy: Coordinated Batching (CB) and Macro-batch Streaming (MS) as Figure 12 depicts. Through CB, the input queries will first be clustered based on their input length and then grouped based on their shared operator. This is to make sure the same sequence length of queries will be executed without wasting padding. MS strategy will take the grouped

queries after coordinated batching and the theoretical latency for different operators as well as the system modeling parameters to generate the best execution order.

The other example design is DLoRA Wu et al. (2024a), which introduces a system that improves the efficiency of serving low-rank adaptation (LoRA) models for large language models (LLMs) by dynamically managing the merging and unmerging of LoRA adapters and the migration of requests across worker replicas. This dynamic orchestration addresses the challenges of high memory footprints, low GPU utilization, and load imbalance caused by variable input and output lengths in traditional LLM serving systems. dLoRA's novel approaches, including a credit-based batching algorithm and a request-adapter co-migration algorithm, significantly enhance throughput.

## 6.3 Distributed PEFT Training Frameworks

We already know that fine-tuning LLM for downstream tasks is challenging for two reasons: dual privacy concerns between cloud server and data owner, and issues with computational resources and efficiency. Firstly, the privacy of both parties is at risk: the weights of large models are often proprietary and not made public. Sharing data with model owners for fine-tuning can lead to data privacy concerns while providing model weights to data proprietors could compromise the ownership of proprietary models. Secondly, even if downstream users have access to pre-trained weights, the stringent hardware requirements make transfer learning impractical for most end users.

To resolve these two issues, **DLoRA** (Gao & Zhang, 2024) presents a distributed PEFT framework. During the PEFT process, the backbone LLM is executed in the cloud servers while the PEFT modules are trained entirely within the user devices. DLoRA scheme is depicted in Figure 10(a).

Similarly, **Offsite-Tuning** (Xiao et al., 2023a) presents a privacy-preserving and efficient transfer learning framework that enables foundational models to adapt to downstream tasks without the need to access the complete model weights. The key insight of Offsite-Tuning is the cloud provider sends an adapter and an emulator to the data proprietor. Then, with the assistance of the emulator, the data proprietor fine-tunes the adapter. The fine-tuned adapter is then sent back to the cloud side, which integrates it into the complete model, creating a fine-tuned foundational model for downstream users. Offsite-Tuning safeguards the privacy of data proprietors since they do not need to share their training data directly. It also protects the foundational model owners, as the complete model weights are not shared, and the emulator provided is lossy, with significantly degraded performance. Compared to existing fine-tuning methods that require access to the full model weights, Offsite-Tuning is more resource-efficient because it allows for fine-tuning through a compressed emulator without needing the complete model.

## 6.4 Parallel PEFT Training Frameworks

Unlike the PEFT query serving system, which aims to accommodate flexible multi-PEFT algorithms, **Punica** (Chen et al., 2023b) focuses solely on facilitating multiple-LoRA blocks for various tasks. Designing multiple PEFT training systems presents key challenges in two main aspects:

- Efficient concurrent execution of multiple PEFT models with the same LLM backbone.

- Designing an efficient system for multi-tenant serving with different LLM backbones.

**Efficient kernel design** Punica addresses the first challenge by using existing matrix multiplication for the backbone computation and introducing a new CUDA kernel, Segmented Gather Matrix-Vector Multiplication (SGMV), for adding the PEFT add-ons to the backbone computation in a batched manner. This kernel parallelizes the feature-weight multiplication for different requests in the batch and groups requests corresponding to the same PEFT model to increase operational intensity and use GPU Tensor Cores for acceleration.

The second challenge is beyond the computational cost, designing an efficient system architecture that can effectively serve multi-tenant PEFT model workloads on the smallest set of GPUs possible while occupying

the least amount of GPU resources is another significant challenge. Punica addresses this by scheduling user requests to active GPUs that already serve or train PEFT models, thereby improving GPU utilization. For older requests, Punica periodically migrates them to consolidate workloads, thus freeing up GPU resources for new requests.

**Multi-Tenant PEFT design**   Designing an efficient system for the multi-tenant PEFT model serving in the Punica framework focuses on addressing several key challenges to maximize hardware utilization and minimize resource consumption. The system aims to consolidate multi-tenant LoRA serving workloads onto the smallest set of GPUs possible. This consolidation is achieved through strategic scheduling of user requests to active GPUs that are already serving or training LoRA models, thereby improving GPU utilization. For older requests, Punica periodically migrates them to consolidate workloads further, thus freeing up GPU resources for new requests. It incorporates on-demand loading of LoRA model weights, which introduces only millisecond-level latency. This feature provides Punica with the flexibility to dynamically consolidate user requests to a small set of GPUs, without being constrained by the specific LoRA models already running on those GPUs. Besides that, Punica identifies that the decode stage is a predominant factor in the cost of model serving, Punica's design primarily focuses on optimizing decode stage performance. Other aspects of model serving leverage straightforward techniques, such as on-demand loading of LoRA model weights, to efficiently manage resource utilization.

# 7   Conclusion and Future Directions

In the current era dominated by large models and large datasets, PEFT stands out as a highly attractive method for efficiently adapting models to downstream tasks. This technique gains its appeal by addressing the significant challenges posed by traditional full-model fine-tuning, which often places substantial computational and data demands. This survey offers a comprehensive examination of the most recent advancements in PEFT, including algorithmic design, computational efficiency, application scenarios, and system implementation for PEFT. It offers a comprehensive taxonomy and explanation that serves as an excellent guidance and knowledge base, which enables readers of various levels and disciplines to swiftly grasp the core concepts of PEFT.

For further research on PEFT, we propose a series of possible directions from both algorithm and system perspectives, hoping to inspire more researchers to engage in further studies in these areas.

## 7.1   Simplify hyperparameter tuning

The effectiveness of PEFT is often sensitive to its hyperparameters, such as the bottleneck dimension of the adapter, the rank of LoRA, and the arrangement of various additive PEFT layers. Manually tuning these hyperparameters will cost lots of effort. Therefore, future efforts could focus on developing methods that are less dependent on manual tuning of these parameters, or automatically find the optimal configuration settings. Several studies (Valipour et al., 2022; Zhang et al., 2023e; Ding et al., 2023a; Chen et al., 2023a; Zhang et al., 2022b; Zhou et al., 2023) have started to address this issue, but there's a need for more simple and efficient solutions optimizing these hyperparameters.

## 7.2   Establish a unified benchmark

Despite the existence of libraries like HuggingFace's PEFT (Mangrulkar et al., 2022) and AdapterHub (Poth et al., 2023), a comprehensive benchmark for PEFT is still lacking. This gap hinders the ability to fairly compare the performance and efficiency of different PEFT approaches. A well-accepted, up-to-date benchmark akin to MMDetection (Chen et al., 2019) for object detection would enable researchers to validate their methods against a standard set of tasks and metrics, fostering innovation and collaboration within the community.

### 7.3 Enhance training efficiency

The presumed parameter efficiency of PEFT is not always consistent with computational and memory savings during training. Given that trainable parameters are intertwined within the pre-trained model's architecture, computing and storing activations and gradients for the full model often become necessary during fine-tuning. This oversight calls for a rethinking of what constitutes efficiency. As outlined in Section 4, potential solutions lie in the integration of model compression techniques such as pruning and quantization, alongside innovations specifically designed to optimize memory during PEFT tuning (Zhang et al., 2023g). Further research into enhancing the computational efficiency of PEFT methodologies is imperative.

### 7.4 Explore scaling laws

The design and effectiveness of PEFT methods originally developed for smaller Transformer models do not necessarily scale with larger models. As the size of foundation models increases, identifying and adapting PEFT strategies that remain effective is crucial. This investigation will aid in customizing PEFT methodologies to suit the evolving landscape of large model architectures.

### 7.5 Serve more models and tasks

The rise of large foundation models across various domains presents new opportunities for PEFT. Designing PEFT methods tailored to the unique characteristics of models, such as Sora (Brooks et al., 2024), Mamba (Gu & Dao, 2023), and LVM (Bai et al., 2023), can unlock new application scenarios and opportunities.

### 7.6 Enhancing data privacy

Trusting centralized systems to serve or fine-tune personalized PEFT modules is yet another issue for system developers. Multiple types of inversion attacks (Dosovitskiy & Brox, 2016; He et al., 2019) have been proposed to reconstruct user's data by hijacking the intermediate results. One perspective of future trust-worthy LLM system design involves developing an encryption protocol for both personal data and intermediate training and inference results.

### 7.7 PEFT with model compression

Model compression is one of the most effective ways to make LLM executable on resource-limited devices. Yet, the impact of model compression techniques on the performance of PEFT algorithms running on hardware remains another systemic challenge. Common compression techniques such as quantization and pruning necessitate dedicated hardware platforms to expedite the process, and building such hardware platforms for compressed models is yet another direction for future research.

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
