# OpenReview forum: "Parameter-Efficient Fine-Tuning for Large Models: A Comprehensive Survey"
_TMLR — Accepted by TMLR_

### Review · Reviewer_H9SJ · 2024-08-14

**Summary Of Contributions:**

This survey paper conducts a comprehensive review on existing parameter-efficient fine-tuning (PEFT) algorithms for large models. The paper revises basic categories of PEFT algorithms, efficient PEFT designs, and introduces literature of applying PEFT in applications such as LLM, ViT, VLA, and diffusion models. The paper also summarizes existing challenges when designing PEFT.

**Audience:**

Yes

**Claims And Evidence:**

Yes

**Requested Changes:**

Please proofread the paper to revise the spelling and grammar.

**Strengths And Weaknesses:**

Strengths: The structure of the paper is clear and the writing is easy to understand. The review is also comprehensive.

Weaknesses: The authors may double check grammars of this paper, e.g., in page 18, there is a misspelling "approace".

---

> ### Author Response · Authors · 2024-09-12
>
> Thank you for your valuable feedback. We have carefully reviewed the grammar and spelling throughout the paper, and we used a grammar checker to ensure accuracy. The revised version of the manuscript has been included in the supplementary materials. For your convenience, the deleted sections are highlighted in red, and the newly added parts are marked in blue.

---

### Review · Reviewer_GqRq · 2024-08-17

**Summary Of Contributions:**

This manuscript provides a literature review of parameter efficient fine-tuning (PEFT) methods, with a focus on fine tuning large language models (LLMs). It provides an ontology of PEFT methods, summarizes the considerations of PEFT design, provides example applications of PEFT outside of LLMs, summarizes considerations for implementing PEFT in systems, and proposes directions of future research.

**Audience:**

Yes

**Claims And Evidence:**

Yes

**Requested Changes:**

1. Run a grammar checker to fix grammar problems.
2. In abstract, avoid hyperbole: "This survey serves as an indispensable resource"
3. Section 3.1.2: "researchers directly append" --> "researchers directly prepend"
4. Section 4.1: "When we look at autoregressive characteristic, it becomes a major challenge in designing an inference system." This is unclear.

**Strengths And Weaknesses:**

Strengths:
- Organization makes sense.
- Ontology makes sense.
- Figures are clear and helpful.
- This is an extremely active area of research, and this manuscript provides a useful high-level overview of the field.

Weaknesses:
- Grammar is poor.
- The word choice is poor in some parts. Some sentences are vague.
- The review skims a lot of papers, and does a reasonable job in organizing the work, but there isn't much in the way of synthesis.

---

> ### Author Response · Authors · 2024-09-12
>
> Thank you for your detailed feedback. We have addressed the points you raised as follows:
>
> 1. We ran a grammar checker to correct any issues throughout the paper.
>
> 2. In the abstract, we revised the wording from "indispensable" to "valuable" to avoid hyperbole.
>
> 3. The correction in Section 3.1.2 has been made, changing "append" to "prepend" as suggested.
>
> 4. For Section 4.1, we have reworked the entire section to improve clarity. We also added more references to support the discussion and resolve any ambiguity.
>
> The revised version of the manuscript has been included in the supplementary materials. For your convenience, the deleted sections are highlighted in red, and the newly added parts are marked in blue.

---

### Review · Reviewer_yYG7 · 2024-08-29

**Summary Of Contributions:**

The paper provides a comprehensive overview of Parameter Efficient Fine-Tuning (PEFT) algorithms for adapting large pre-trained models to specific downstream tasks while minimizing the number of additional parameters and computational resources required.
The key contributions of the paper are:

1. It examines the performance and computational overhead of various PEFT algorithms. This helps understand the trade-offs between fine-tuning efficiency and model performance.
2. It provides an overview of applications developed using different PEFT algorithms. This illustrates the practical utility and versatility of PEFT techniques across domains.
3. It discusses common techniques employed to mitigate the computational costs of PEFT.
4. Beyond just the algorithms, it also examines real-world training and serving system designs to investigate the implementation costs of different PEFT approaches, which are crucial for practical deployment.

**Audience:**

Yes

**Claims And Evidence:**

Yes

**Requested Changes:**

To strengthen the work:

1. Comprehensive benchmarking: To provide a more thorough evaluation, I recommend adding benchmarks for different key algorithms (e.g., various forms of PEFT methods) using diverse metrics. Additionally, benchmarking different training and serving systems for PEFT workloads and comparing their metrics would be beneficial. A more detailed discussion of existing evaluation practices would also enhance the work.

2. Correction in Section 6.4: I noticed that S-Lora and Punica are actually Lora serving frameworks, whereas the section title is "Parallel PEFT Training Frameworks". I suggest that the authors correct this inconsistency to ensure clarity and accuracy.

3. Abstraction of challenges and design principles: In the PEFT system parts, I recommend that the authors abstract the challenges and key design principles for PEFT workloads instead of simply listing related works and their methods. This would provide a more insightful and structured discussion, allowing readers to better understand the underlying complexities and design considerations for PEFT systems.

**Strengths And Weaknesses:**

Strengths:

The paper provides a comprehensive review of Parameter-Efficient Fine-Tuning (PEFT) algorithms, covering additive, selective, reparameterized, and hybrid approaches. Additionally, the authors discuss efficient optimization techniques for deploying PEFT models, as well as some existing system examples. Overall, this survey serves as a valuable resource for understanding and utilizing PEFT techniques in the era of large pre-trained models.

Weakness:

1. Lack of clarity on KV-cache management: In the efficient PEFT design section, it is unclear how to specifically conduct KV-cache management for PEFT tasks. Providing more detailed guidance on this aspect would enhance the practicality of this part.
2. Limited coverage of recent works: The PEFT serving system parts could be improved by including other recent works, such as dLora (OSDI 2024). This would ensure that the survey remains up-to-date and comprehensive.
3. Absence of benchmarks: The survey lacks related benchmarks regarding both algorithm and system optimization. Including such benchmarks would provide a more thorough evaluation of the PEFT techniques and systems, allowing readers to better understand their performance and trade-offs.

---

> ### Author Response · Authors · 2024-09-12
>
> Thank you for your detailed feedback. We have made several revisions to address the concerns you raised:
>
> 1. We have added a new section, Section 2.4, that discusses benchmarks. We include several benchmarks across algorithm and system perspective to enable readers to evaluate the performance differences between various PEFT methods.
>
> 2. The section on KV cache management has been enhanced to offer clearer guidance, along with added references to recent papers for further support.
>
> 3. We have incorporated the recent work, dLora (OSDI 2024), to ensure the survey is more comprehensive and up-to-date.
>
> 4. Following your recommendations regarding Sections 6.4 and the system discussion, we corrected the inconsistency with the S-Lora and Punica frameworks and abstracted the challenges and design principles for PEFT workloads, providing a more structured and insightful discussion.
>
> The revised version of the manuscript has been included in the supplementary materials. For your convenience, the deleted sections are highlighted in red, and the newly added parts are marked in blue.

---

### Decision · Action_Editor_36sw · 2024-10-20

**Recommendation:** Accept with minor revision

**Comment:**

Reviewer GqRq highlighted grammar issues and recommended revising hyperbolic language like "indispensable" to more neutral terms, such as "valuable". Also reviewer H9SJ also mentioned minor spelling and grammar errors that need correction throughout the paper. Please fix these for the final version.

**Audience:**

The topic of this paper is highly relevant to a broad section of TMLR's audience. Parameter-efficient fine-tuning is a critical area of research for those working with large models, particularly in light of the growing computational demands associated with these models. As Reviewer yYG7 noted, the paper provides valuable insights into the practical implementation of PEFT in real-world systems, which is of interest for both researchers (so they can pursue useful research) and practitioners.

**Claims And Evidence:**

The authors present a clear and thorough review of Parameter-Efficient Fine-Tuning (PEFT) algorithms, examining both their algorithmic performance and system-level implementation. As Reviewer yYG7 pointed out, the paper provides "a comprehensive review" that covers various PEFT techniques, trade-offs in computational costs, and applications across different domains. The authors have also responded to feedback by adding benchmarks and addressing technical concerns, thus ensuring that the claims made are backed by well-documented evidence​. Reviewer GqRq emphasized the clarity and organization of the paper, calling it a "useful high-level overview" of an important and active research area.